There are amendments to this paper

# Atypical function of a centrosomal module in WNT signalling drives contextual cancer cell motility

Yi Luo[1], Miriam Barrios-Rodiles[1], Gagan D. Gupta[1,3], Ying Y. Zhang [1,2], Abiodun A. Ogunjimi[1], Mikhail Bashkurov[1], Johnny M. Tkach[1], Ainsley Q. Underhill[1,2], Liang Zhang[1,4], Mohamed Bourmoum[1], Jeffrey L. Wrana[1,2] & Laurence Pelletier [1,2]

Centrosomes control cell motility, polarity and migration that is thought to be mediated by their microtubule-organizing capacity. Here we demonstrate that WNT signalling drives a distinct form of non-directional cell motility that requires a key centrosome module, but not microtubules or centrosomes. Upon exosome mobilization of PCP-proteins, we show that DVL2 orchestrates recruitment of a CEP192-PLK4/AURKB complex to the cell cortex where PLK4/AURKB act redundantly to drive protrusive activity and cell motility. This is mediated by coordination of formin-dependent actin remodelling through displacement of cortically localized DAAM1 for DAAM2. Furthermore, abnormal expression of *PLK4*, *AURKB* and *DAAM1* is associated with poor outcomes in breast and bladder cancers. Thus, a centrosomal module plays an atypical function in WNT signalling and actin nucleation that is critical for cancer cell motility and is associated with more aggressive cancers. These studies have broad implications in how contextual signalling controls distinct modes of cell migration.

[1] Lunenfeld-Tanenbaum Research Institute, Mount Sinai Hospital, Toronto, ON M5G 1X5, Canada. [2] Department of Molecular Genetics, University of Toronto, Toronto, ON M5S 1A8, Canada. [3]Present address: Department of Chemistry and Biology, Ryerson University, Toronto, ON M5B 2K3, Canada. [4]Present address: Department of Biomedical Sciences, City University of Hong Kong, Hong Kong, China. Correspondence and requests for materials should be addressed to L.P. (email: pelletier@lunenfeld.ca)

Cell motility and invasive behaviour of tumour cells is a key mechanism underlying cancer metastasis. To invade the stroma, cancer cells activate signalling pathways that control cytoskeletal reorganisation, cell matrix interactions and cell–cell junction turnover[1]. Dynamic remodelling of the cortical actin cytoskeleton into a filamentous configuration is a critical step that facilitates cell elongation and migration[2]. This generates actin-rich membrane projections such as lamellipodia, filopodia, blebs and invadopodia that in turn promote cell motility, invasion and extracellular matrix (ECM) degradation[3]. At the molecular level, spatiotemporal control of actin dynamics is mediated by signalling networks that act on Rho GTPase family members, profilin, ADF/cofilin, Wiskott–Aldrich syndrome protein (WASP), WASP-family verprolin-homologous protein (WAVE) family proteins, actin-related protein 2/3 (ARP2/3) complex and formins[4,5]. The formation of lamellipodia is predominantly mediated by the ARP2/3 complex, which induces actin meshworks in protrusions, and RAC, RHOA and CDC42 GTPases that are key regulators of cytoskeletal organisation[6]. Filopodia, another important protrusive element can either emerge from lamellipodia or can form independently via formin-dependent regulation of actin polymerisation[7].

Microenvironmental signals emanating from cancer-associated fibroblasts (CAFs) are important for controlling metastasis[8]. We previously demonstrated that exosomes secreted from activated fibroblasts and CAFs stimulate breast cancer cell (BCC) protrusive activity, motility, and metastasis through WNT-planar cell polarity (PCP) signalling[9]. In the receiving BCC, exosomes mobilise autocrine WNT11 to stimulate its association with FZD6, and asymmetric accumulation of FZD-DVL in protrusions, while VANGL and PRICKLE localise to the flanking lateral non-protrusive cortical regions[9], where they associate with ARHGAP21/23 to restrict RHOA activity, thereby regulating actomyosin activation and focal adhesion (FA) dynamics[10]. However, the exact molecular mechanisms that initiate the reorganisation of the actin cytoskeleton in response to exosomes and how this process controls cell migration speed remain unclear.

The centrosome is the primary microtubule-organising centre (MTOC) in animal cells, and it is important for many cellular processes, including cell motility, polarity, division and signalling[11]. Centrioles are duplicated once per cell cycle where the key regulating kinase, PLK4, is recruited to the mother centrioles via CEP192 and CEP152 at the G1-S transition[11]. Impaired centriole duplication results in numerical centrosomal aberrations, which is a common feature of many human tumours[12]. As such, PLK4 activity is tightly regulated through trans-autophosphorylation, which generates a phosphodegron targeted by the SCF E3 ubiquitin ligase complex[13,14]. Centrosome aberrations can alter the motile properties of cells, for example, PLK4-driven centrosome amplification increases RAC1 activity at the cortex and promotes cell invasion[15]. During directional cell migration, the centrosome is positioned at the front of the cell between the nucleus and the leading edge, where it nucleates MTs and serves as a hub for directional transport of membrane components and signalling molecules. Regulation of cell migration by centrosomes is typically explored in the context of MT network organisation; however, centrosomes can also act as actin nucleation sites through the recruitment of the actin nucleation-promoting factor WASH and ARP2/3 by the centriolar satellite protein PCM1[16]. Furthermore, PLK4 was proposed to participate in cell migration independently of its role in controlling the MT landscape by stimulating directional cell migration through ARP2/3-mediated actin rearrangement, independent of RAC1 and CDC42[17,18]. However, a mechanistic understanding of how centrosomes

exert contextual control of actin remodelling and cell motility is thus far lacking.

Here, we reveal an atypical function for a centrosomal module in exosome-WNT signalling stimulated cancer cell motility. We show that while exosome-induced cancer cell motility requires neither centrosomes nor the MT network, it is critically dependent on CEP192, the CEP192 interacting kinase PLK4, and the Chromosomal Passenger Complex (CPC) protein Aurora Kinase B (AURKB). In response to ACM stimulation, the PCP protein DVL2 initiates recruitment of a CEP192-PLK4-AURKB module to protrusions. This promotes CEP192 localisation to the cell cortex, PLK4 stability and AURKB nuclear exit. In turn, PLK4 and AURKB promote switching of the formins DAAM1 for DAAM2 in cell protrusions, which then drives actin re-organisation and cell migration. We further show that key components of this axis are aberrantly expressed in the most aggressive forms of breast and bladder cancers. Globally, our studies unravel an unexpected function for a centrosomal module in WNT signalling and highlight how the contextual activation of this signalling axis by microenvironmental cues regulates the motility and invasive properties of cancer cells.

## Results

**Exosomes stimulate non-directional cell motility, independent of centrosomes and the MT network.** We previously reported that the L cell fibroblast active-conditioned-medium (ACM) or purified exosomes mobilise WNT signalling in a variety of BCC lines to stimulate protrusive activity, cell shape dynamics and motility[9] (e.g., MDA-MB-231, Supplementary Movie 1). Centrosomes have well-described functions in directional cell migration (e.g., wound healing), so here we investigated centrosome function in exosome/WNT signalling-mediated cell motility by using the PLK4 inhibitor centrinone[19], which causes progressive loss of centrioles after multiple rounds of cell division (Fig. 1a). After 5 days of exposure to centrinone, ~85% of MDA-MB-231 cells had lost centrosomes (Fig. 1b, c), but strikingly their motility in response to ACM was unaltered (Fig. 1d). To account for the small proportion of cells that might contain centrosomes after centrinone treatment, we generated a cell line stably expressing GFP-CETN2 to mark centrioles. Cells treated with or without centrinone were mixed after 5 days of incubation and stimulated with ACM (Supplementary Fig. 1a and Supplementary Movie 2). Cells with or without centrosomes both responded to ACM stimulation by continuously assembling long and dynamic protrusions, and exhibited near identical velocity (Supplementary Fig. 1a). Three-dimensional structured-illumination (3D-SIM) imaging further revealed that without ACM stimulation, cells were rounded with continuous, plasma membrane proximal actin and MT bundles (Fig. 1e). In stark contrast, ACM-stimulated cells, with or without centrosomes, formed multiple long protrusions and re-organised their actin cytoskeleton into short filamentous networks (Fig. 1e). Together, these data show that centrosomes are not required for ACM-stimulated BCC protrusive activity and motility.

The dispensability of centrosomes in ACM-stimulated motility contrasts their requirement in directional migration, such as wound-healing assay, where centrosome loss suppresses wound closure[20] (Supplementary Figs. 1b, c). Our previous work implied that ACM-stimulated BCC migration displayed random walk properties[10]. To confirm this observation, we measured the mean-square displacement (MSD) of cell migration traces, which is a common measure of the spatial extent of motion. Our data revealed that exosome-induced non-directional movements indeed display random walk properties (when the MSD and time display a linear relationship ($R^2 = 1$) this is indicative of the

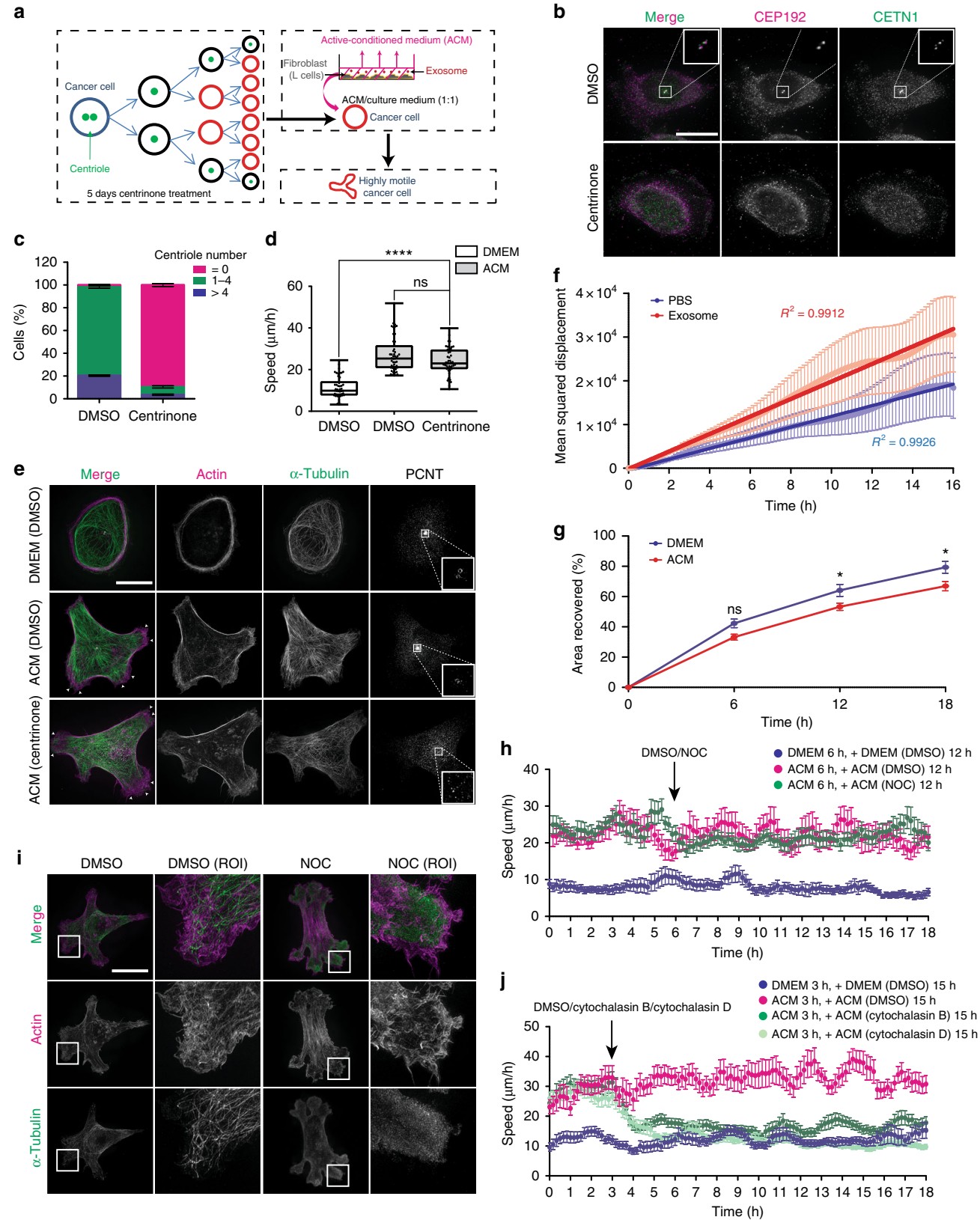

total random movement) (Fig. 1f; Supplementary 1d)[9,10]. We therefore hypothesised that ACM-stimulated BCC migration reflects an intrinsically different mechanism of cell motility compared with directional migration. Accordingly, we observed that ACM significantly reduced closure rates (~13% at 18 h) in wound-healing assays, which suggests that ACM-induced non-

directional migration might compete with cellular pathways controlling directional migration[21] (Fig. 1g; Supplementary Fig. 1e). Surprisingly, disrupting the MT network, which is crucial in directional migration[20], with nocodazole (Supplementary Fig. 1f), did not inhibit ACM-induced motility (Fig. 1h). Furthermore, actin and MT network rearrangement at the cell

**Fig. 1** Exosomes stimulate actin-based non-directional migration in a centrosome and MT-independent manner. **a** Experimental schematic. MDA-MB-231 cells were treated with centrinone for 5 days to induce centrosome loss during division. Centrosome-less cells were treated with ACM. The protrusive activity and motility of individual cells were imaged and tracked. **b** MDA-MB-231 cells were treated with DMSO or centrinone for 5 days. CETN1 (centrioles) and CEP192 (centrosomes) staining is shown. Bar = 20 μm. **c** Percentage of cells from (**b**) with > 4 centrioles, $1 \leq$ centrioles $\leq 4$ and 0 centrioles ($n = 3$ independent experiments analysing at least 100 cells per each replicate). **d** MDA-MB-231 cells were treated with DMSO or centrinone for 5 days, followed by overnight incubation with the DMEM or ACM. Cell motility was measured as described in Methods and plotted as box and whiskers. Boxes represent median and 25th to 75th percentiles, whiskers the minimum and maximum values with each individual cell value superimposed. The data were compared with one-way ANOVA Kruskal–Wallis test and post-tested with Dunn's multiple comparison test. Arrow indicates the control bar used for comparison (****$p < 0.0001$; $n = 4$, at least 50 cells were tracked per condition). **e** MDA-MB-231 cells from (**d**) were stained for α-tubulin, actin (phalloidin) and centrosomes (PCNT). Arrowhead indicates cell protrusion. Bar = 20 μm, inset box 3.88 × 3.88 μm ($n = 3$). **f** Mean-squared displacement (MSD) of cells stimulated with PBS (light blue) or purified exosomes (light red). Linear regression analysis of MSD was preformed (dark blue line is PBS, dark red line is purified exosomes). MSD $R^2 = 1$ indicates a total random movement ($n = 3$, at least 40 cells were tracked per condition). **g** Quantitative data from wound-healing assay on MDA-MB-231 cells treated overnight with DMEM or ACM are shown in Supplementary Fig. 1e. Graph shows mean area recovered. The data were analysed with two-way ANOVA post-tested with Bonferroni test (*$p < 0.05$; $n = 3$, at least 30 regions were measured per condition). **h** MDA-MB-231 cells were stimulated with DMEM or ACM overnight. Cell motility was tracked for 6 h. Then, the same cells were further tracked after being treated with DMSO or Nocodazole for 12 h. Running average speed was plotted ($n = 3$, at least 40 cells were tracked per condition). **i** MDA-MB-231 cells were stimulated with ACM overnight. DMSO or NOC were added for the last 2 h of stimulation. Actin (red) and α-tubulin (green) staining are shown. Region of interest (ROI) shows the enlarged area of the selected cell protrusion. Bar = 20 μm ($n = 3$) **j** MDA-MB-231 cells were stimulated with DMEM or ACM overnight. Cell motility was tracked for 3 h, then cells were treated with DMSO, cytochalasin B or cytochalasin D and further tracked for 15 h. Running average speed was plotted ($n = 3$, at least 40 cells were tracked per condition). The data from **c, f, g, h** and **j** are plotted as mean ± s.e.m.

cortex occurred in a similar fashion than control-stimulated cells (Fig. 1i). In stark contrast, disruption of the actin network using cytochalasin B or cytochalasin D caused strong inhibition of cell motility (Fig. 1j). Overall, these results show that ACM-stimulated non-directional migration is actin dependent, but independent of intact centrosomes and the MT network. Furthermore, they suggest that ACM-WNT signalling driven motility response to extrinsic cues is distinct from conventional directional cell motility pathways.

**Centrosomal components are required for exosome-stimulated cancer cell motility**. The centriole duplication/maturation factor CEP192 exerts control on directed cell motility through increased MT nucleation at centrosomes[15,22]. Surprisingly, in contrast to centrosome loss, siRNA depletion of CEP192 inhibited ACM-induced protrusive activity (Fig. 2a; Supplementary Fig. 2a, b), and exosome-induced cell motility (Supplementary Fig. 2d). Moreover, in 3D culture, ACM induced protrusion formation in a highly dynamic manner in siCtrl-treated cells, in contrast to siCEP192-treated cells that were ACM stimulated (Supplementary Fig. 2c, Supplementary Movie 3). Furthermore, CEP192 depletion phenocopied knockdown of the PCP protein, PRICKLE1 (Fig. 2a)[9,10], leading us to hypothesise that CEP192 may affect WNT signalling in response to ACM in a centrosome-independent manner. To confirm this, we depleted centrosomes using centrinone, and then knocked-down CEP192 prior to ACM stimulation (Fig. 2b). In contrast to control RNAi-treated cells, CEP192 depletion inhibited cell motility in the absence of centrosomes (Fig. 2b). We next evaluated the requirement of CEP192-mediated MT nucleation in the context of ACM-stimulated motility. CEP192 is necessary for the recruitment of NEDD1 to centrosomes, which in turn recruits gamma-tubulin ring complexes and subsequent MT nucleation[23,24]. To assess MT regrowth activity, CEP192 or NEDD1 siRNA-treated MDA-MB-231 cells (Supplementary Fig. 2e) were incubated on ice to disassemble the MT network before shifting back to 37 ºC to allow regrowth. As expected, knockdown of CEP192 or NEDD1 significantly reduced MT nucleation from centrosomes (Fig. 2c, d). However, unlike CEP192 RNAi treatment, knockdown of NEDD1 did not inhibit ACM-induced motility (Fig. 2e). Furthermore, to exclude the possibility that CEP192 down-regulation might alter exosome uptake, modification and

secretion, we determined the levels of endogenous WNT11 loaded into exosomes upon their uptake into MDA-MB-231 cells (Supplementary Fig. 2f). Our results indicate that control and CEP192 RNAi-treated BCCs process similar amounts of CD81-positive exosomes (released by L cell fibroblasts) that are loaded with endogenous WNT11 (Supplementary Fig. 2g). Thus, CEP192 is required for ACM-induced BCC motility in a manner independent of centrosomes and their MT nucleation activity.

**PLK4-AURKB/C redundantly regulate ACM-stimulated cell motility**. Elevated PLK4 expression is proposed to stimulate the invasive properties of breast cancer cells via CEP192-mediated MT nucleation at centrosomes[15]. Although centrinone, a highly selective PLK4 inhibitor, did not affect motility (see Fig. 1a–e) in the course of these studies, we observed that another PLK4 inhibitor, CFI-400945[25] potently inhibited ACM-induced movement (Fig. 3a) and reduced protrusive activity (Fig. 3b). Since CFI-400945 also inhibits AURKB and AURKC[25], we reasoned that PLK4 and AURKB and/or C may act redundantly to regulate BCC motility in response to exosomes. To test this, we used AZD1152 (AZD), an AURKB/C inhibitor together with centrinone (or centrinone B), and found that only co-inhibition of AURKB/C and PLK4 prevented ACM-induced BCC migration (Fig. 3c) and suppressed protrusive activity (Fig. 3d). We then targeted either PLK4 or the Aurora kinase family members (A, B and C) using siRNA. Knockdown of PLK4, single AURKs, or the complement of AURKs had no effect on protrusive activity and motility (Supplementary Fig. 3a–d). However, when PLK4 and both AURKB and AURKC were targeted, ACM-induced protrusive activity and cell motility were inhibited (Supplementary Fig. 3a, b).

We next exploited PLK4 and AURKB/C small-molecule inhibitors to assess the reversibility of inhibition by performing wash-out experiments, where we tracked ACM-stimulated cells treated with centrinone and AZD for 16 h before and after washout (Fig. 3e). While control cells remained highly motile throughout, addition of centrinone and AZD suppressed motility that was rapidly recovered after washout (Fig. 3e). We noticed that dual inhibition of PLK4 and AURKB required about 14 h to fully suppress ACM-induced cell motility, with a half-life decay at 1.7 h. To rule out that this rather slow inhibition of motility

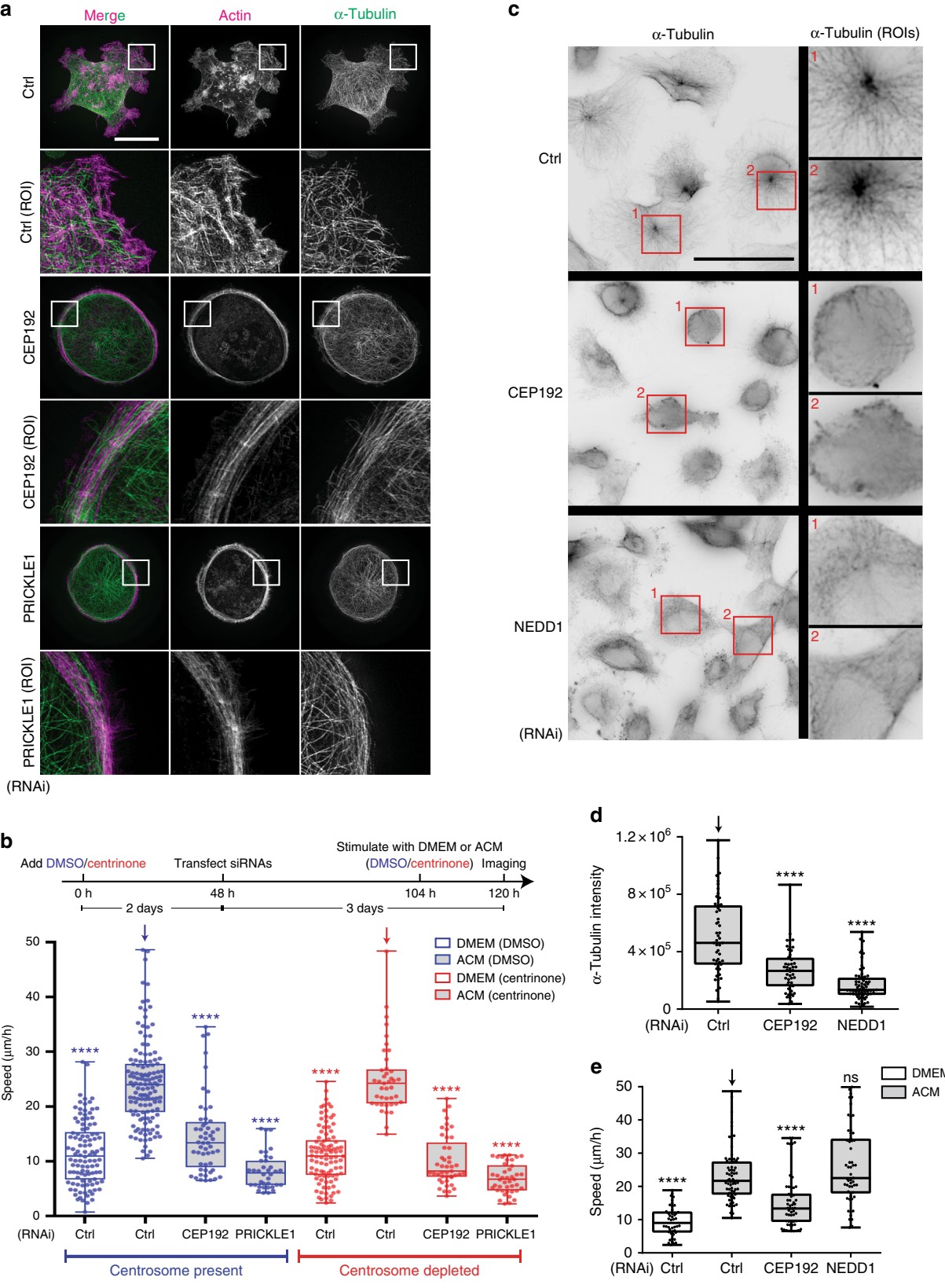

occurred indirectly through the inhibition of AURKB/PLK4-controlled transcription, we therefore tested the effect of inhibiting transcription using actinomycin D on cell motility. Although actinomycin D treatment did inhibit ACM-induced cell motility, the inhibition kinetics was much slower (about 5.5 h for a half-life decay, Supplementary Fig. 3e, f). Although we cannot

exclude the possibility at this stage that AURKB/PLK4 also regulate cell motility via transcription, our preferred hypothesis, considering that AURKB/PLK4 localisation and their activation is induced upon ACM stimulation (see below), is that these kinases likely have direct roles in actin rearrangement that require further investigation.

**Fig. 2** Non-centrosomal pool of CEP192 controls ACM-induced motility. **a** MDA-MB-231 cells were transfected with Control (Ctrl), CEP192 or PRICKLE1 siRNAs for 72 h and then stimulated with ACM overnight. Cells were then stained for α-tubulin and actin. Region of Interest (ROI) panels show the enlarged box area at the cell cortex or protrusions. Bar = 20 μm (n = 3). **b** MDA-MB-231 cells were treated with DMSO or centrinone for 5 days. Two days after drug addition, cells were transfected with the indicated siRNAs and further incubated for 3 days. Cells were then treated with DMEM or ACM overnight at the end of the 5-day incubation and time-lapse images were taken to track cell motility (****$p < 0.0001$; n = 3, at least 50 cells were measured per condition). **c** MDA-MB-231 cells were transfected with siRNAs for 72 h. Next, cells were put on ice for 30 min, followed by recovery at 37 °C for 20 s. Cells were then fixed and stained for PCNT and α-tubulin. ROI panels show enlarged area at microtubule-organising centre (MTOC). Bar = 50 μm. **d** α-tubulin intensity at MTOC (mask defined by PCNT signal) in cells from (**c**) was measured from three independent experiments. (****$p < 0.0001$; n = 3, at least 50 cells were measured per condition). **e** Cell motility was measured as average speed in MDA-MB-231 cells transfected for 72 h with the indicated siRNAs and then treated with DMEM or ACM overnight (****$p < 0.001$; n = 3, at least 50 cells were tracked per condition). All the data were plotted as box and whiskers. Boxes represent median and 25th to 75th percentiles, whiskers the minimum and maximum values with each individual cell value superimposed. The data were compared with one-way ANOVA Kruskal–Wallis test and post-tested with Dunn's multiple comparison test. Arrow indicates the control bar used for comparison

Finally, we tested the redundant role of PLK4 and AURKB/C in stimulating ACM-induced cell migration in other cell lines. ACM induced high motility in mouse (EpRas), and in human (MDA-MB-468) breast cancer cell lines (Fig. 3f), as well as two high-grade bladder cancer cell lines, TCCSUP and T24 (Supplementary Fig. 3g). In all cases, motility was only blocked by co-inhibition of PLK4 and AURKB/C, indicating these kinases are functionally redundant in a variety of cancer contexts. Importantly, analysis of data from 960 breast cancer patients deposited in the TCGA (http://www.cbioportal.org) indicates that elevated expression of *PLK4*, *AURKB* and *AURKC* correlates with reduced survival rates (Supplementary Fig. 3h). Similarly, analysis of expression by RNA-seq in a cohort of 158 bladder cancer patients revealed that elevated expression of *AURKB* and *PLK4* was associated with high-grade disease, in contrast to *GAPDH*, which was expressed similarly in both grades (Supplementary Fig. 3i). Of note, *AURKC* transcripts were barely detectable (Supplementary Fig. 3i). Together, these observations indicate that PLK4 and AURKB/C act redundantly to promote ACM-stimulated cell motility in various cancers and are associated with more aggressive breast and bladder cancers.

**CEP192, PLK4 and AURKB/C associate with the WNT-PCP protein DVL2**. Interference with the CEP192-PLK4-AURKB/C module inhibits ACM-induced protrusive activity and cell motility in a manner analogous to perturbing WNT signalling [9] (e.g. Fig. 2a, b), suggesting they might function in the same pathway. To explore this, we mined our map of the centrosome–cilium interface, which revealed a number of interactions between centrosomal components and PCP proteins [26], and used the automated luminescence-based mammalian interactome (LUMIER) assay [27] to systematically screen interactions between more than 79 WNT-PCP proteins and CEP192, PLK4 and AURKs. This revealed DVL2 as a hub that interacted with all four proteins (Fig. 4a; Supplementary Fig. 4a and Supplementary Table 1). We validated the interactions between DVL2 and CEP192 (Fig. 4b), and with AURKs or PLK4 by co-immunoprecipitation (co-IP, Supplementary Fig. 4b, c). We further confirmed the interaction of endogenous DVL2 with purified AURKB and PLK4 proteins, and showed that bacterially expressed GST-AURKB or GST-PLK4 associated with endogenous DVL2 from BCC whole-cell lysates (Fig. 4c, d). Domain-mapping experiments with DVL2 mutants (Supplementary Fig. 5i) further showed that the interactions between DVL2 and AURKB, PLK4 or CEP192 depend on both the N-terminal and C-terminal halves of DVL2 (Supplementary Fig. 5a–e). Although we carried out more precise domain deletions in DVL2, for instance, ΔDEP shows significant inhibition of DVL2 association with PLK4 and CEP192, the expression/stability of this mutant is so poor (about 5% of the full-length DVL2) that we cannot draw meaningful conclusions. While

performing these studies, we also observed that co-expression of DVL2 with PLK4 led to a strong increase in PLK4 steady-state levels that correlated with their physical interaction (Supplementary Fig. 5b; examined further below). Overall, these data indicate that binding of CEP192/PLK4/AURKB to DVL2 requires regions in both its N- and C-halves.

We next used RNAi to knockdown DVL2, which inhibited ACM/exosome-induced cell motility (Fig. 4e; Supplementary Figs. 5f, g), and induced a rounded cell phenotype without protrusions (Fig. 4f), reminiscent of CEP192 knockdown (Fig. 2a) or double inhibition of PLK4 and AURKB/C (Fig. 3b, d; Supplementary Fig. 3b). To confirm the specificity of the DVL2 RNAi treatment, we depleted DVL2 with an siRNA targeting its 3′-UTR in MDA-MB-231 stably expressing tetracycline (Tet)-inducible wild-type (wt) DVL2 or the N- and C-terminal halves, which failed to interact with either PLK4, AURKB or CEP192 (Supplementary Fig. 5a–e). Adjusting the Tet concentration allowed for expression at similar levels to endogenous DVL2 (Supplementary Fig. 5h). Importantly, after knockdown of endogenous DVL2, Tet-induced DVL2-wt expression rescued ACM-induced motility, while the DVL2 mutants did not (Fig. 4g). Collectively, these data are consistent with the notion that DVL2 regulation of BCC motility is dependent on its interaction with CEP192, PLK4 and AURKB.

**DVL2 orchestrates CEP192, PLK4 and AURKB recruitment to protrusions**. In the WNT signalling pathway, WNT binding to seven-transmembrane FZD receptors leads to recruitment of DVL to the membrane, where it regulates cytoskeletal reorganisation through diverse downstream effectors [28]. FZD recruits DVL to the cell cortex during ACM-stimulated BCC migration [9], which we show here also includes phospho-DVL2 (Supplementary Fig. 6a). We hypothesised that DVL2 might thus recruit CEP192, PLK4 and AURKB/C to the cell cortex and designed an automated algorithm to measure proteins of interest in the cortical region (see the Methods section and Supplementary Fig. 6b). Indeed, CEP192 accumulated at protrusions after ACM stimulation (Fig. 5a, b). Interestingly, centrosomal CEP192 was concomitantly reduced by ACM stimulation (Fig. 5c, d), but not PCNT (Fig. 5c, e), which was not required for ACM-induced motility (Supplementary Fig. 6c). Importantly, DVL2 RNAi blocked the cortical pool of CEP192 and maintained CEP192 at the centrosome similar to non-stimulated cells (Fig. 5a–d). These findings suggest that DVL2 recruits CEP192 to protrusions either from the centrosome or through a cytoplasmic pool which continuously exchanges with the centrosomal pool of CEP192 [24] (Fig. 5f).

We next examined the localisation of active PLK4 phosphorylated at S305 or T170 (pS305 [29]; pT170 [30]) and active AURKB phosphorylated at T232 (pT232 [31]). We observed that ACM

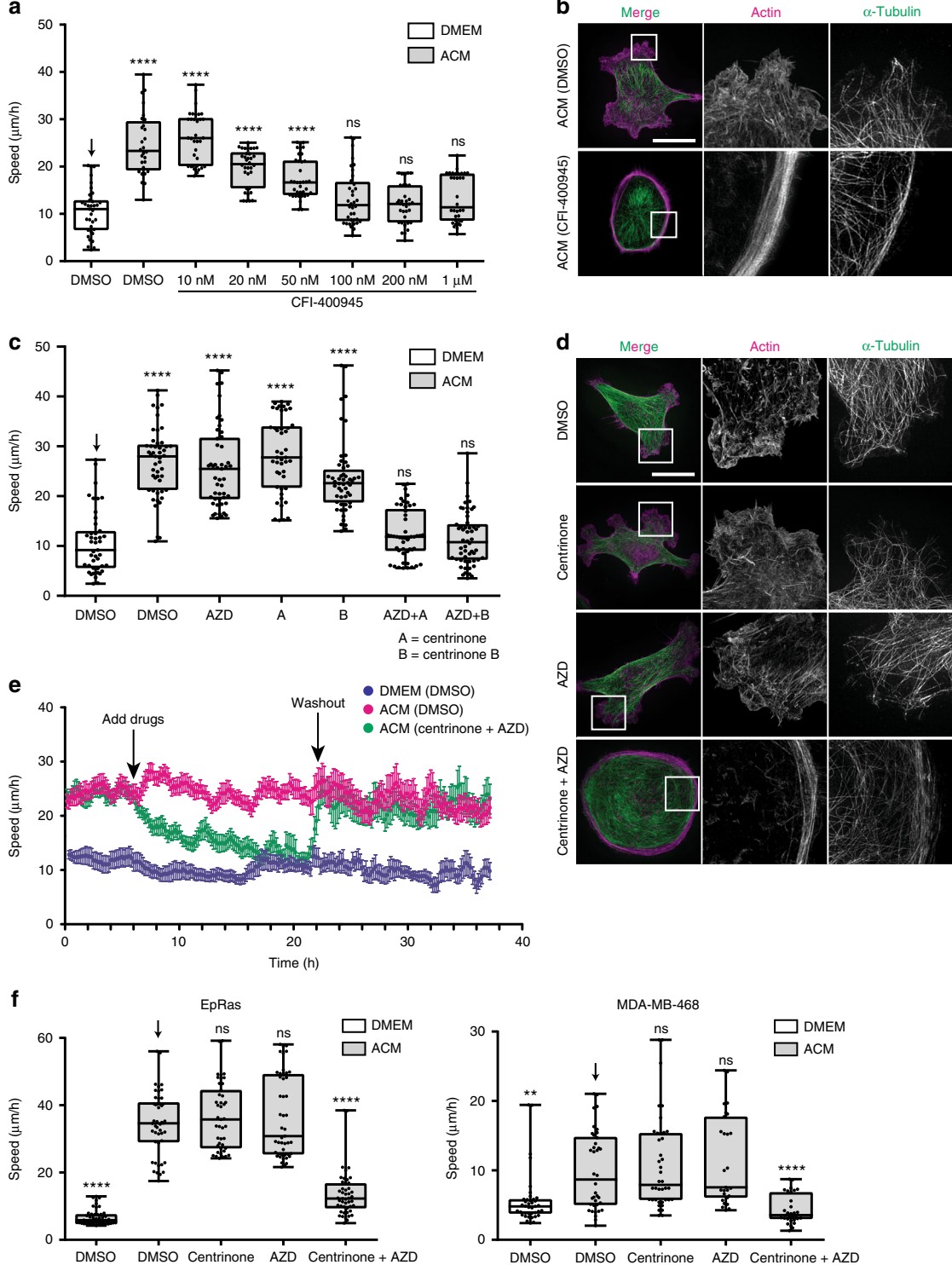

**Fig. 3** PLK4 and AURKB/C redundantly modulate ACM-induced cancer cell motility. **a** MDA-MB-231 cells were treated with DMSO or different concentrations of CFI-400945 (PLK4 and AURKB/C inhibitor), in the presence of DMEM or ACM (****$p < 0.0001$; $n = 3$, at least 50 cells were tracked per condition). **b** MDA-MB-231 cells were treated with DMSO or 100 nM CFI-400945 in the presence of ACM stimulation. Cells were stained for α-tubulin and actin, and enlarged images showed a cortical region of the cell. Bar = 20 μm ($n = 3$). **c** MDA-MB-231 cells were treated with centrinone, centrinone B (PLK4 inhibitors), AZD (AURKB/C inhibitor) or their combination, in the presence of DMEM or ACM overnight (***$p < 0.0001$; $n = 3$, at least 50 cells were tracked per condition). **d** Cells from (**c**) were stained for α-tubulin and actin, and cortical regions were enlarged on right side panels. Bar = 20 μm. **e** Cells were incubated with DMEM or ACM overnight. At 6 h of taking time-lapse images, DMSO or inhibitors were added to cells and incubation continued for 14 h more, then DMSO or inhibitors were washed out and cells were further incubated in DMEM or ACM for another 16 h. Running average speed was plotted as mean ± s.e.m. ($n = 4$, at least 50 cells were tracked per condition) **f** Breast cancer cell lines EpRas and MDA-MB-468 were treated with DMEM or ACM in the presence of DMSO or inhibitors (**$p < 0.01$, ****$p < 0.0001$; $n = 3$, at least 40 cells were tracked per condition). All the data were plotted as box and whiskers. Boxes represent median and 25th to 75th percentiles, whiskers the minimum and maximum values with each individual cell value superimposed. The data were compared with one-way ANOVA Kruskal–Wallis test and post-tested with Dunn's multiple comparison test. Arrow indicates the control bar used for comparison

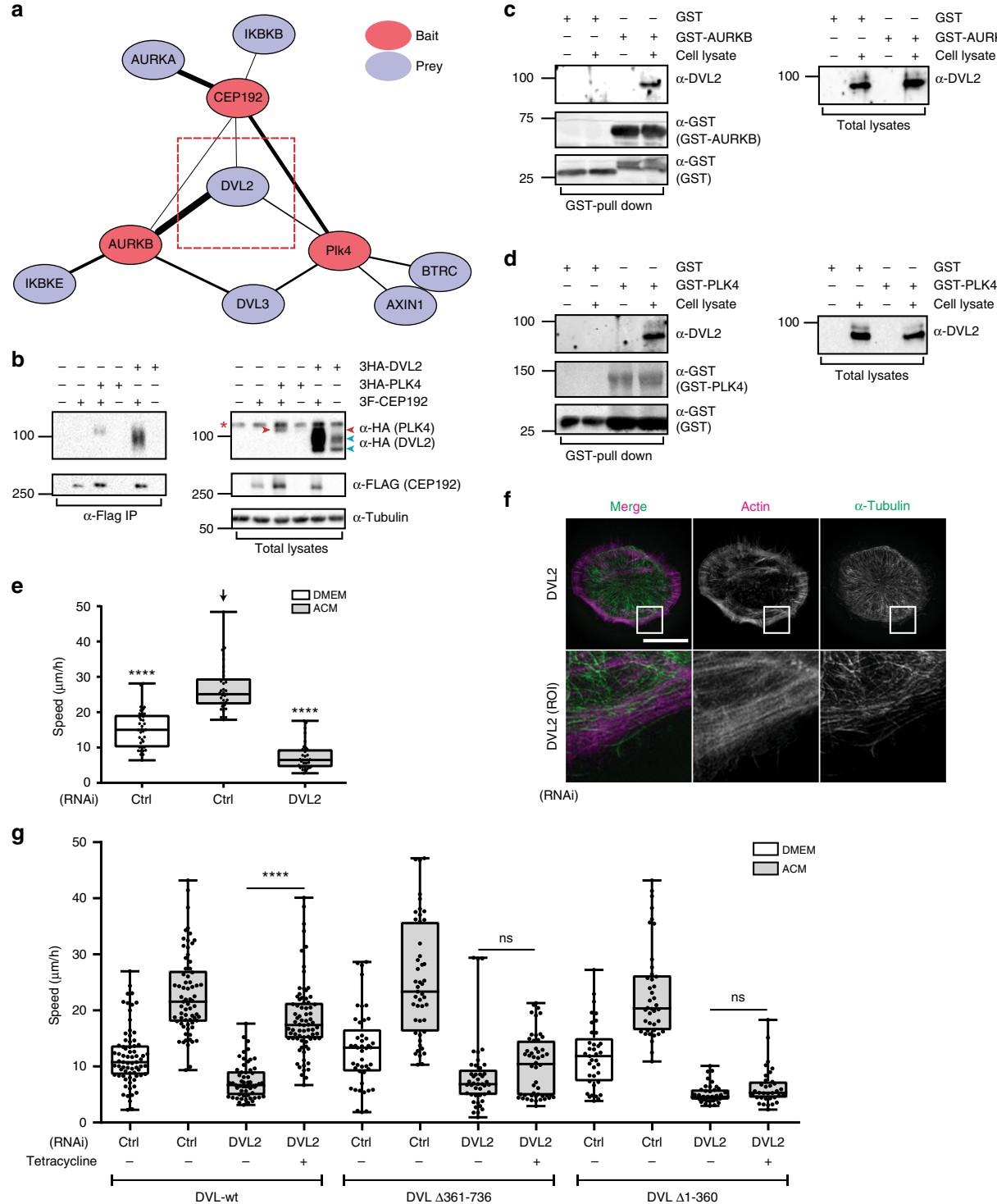

induced accumulation of PLK4pS305 (Fig. 6a, b), PLK4pT170 (Supplementary Fig. 6d, e) and AURKBpT232 at protrusions (Fig. 6c, d). Consistent with their kinase specificities, centrinone inhibited the appearance of PLK4pS305 at the protrusions (Fig. 6a, b), but not AURKBpT232 (Figs. 6c, d, f), while AZD inhibited AURKBpT232 accumulation (Fig. 6c, d), but not PLK4pS305 (Fig. 6a, b, e). Knockdown of PLK4 (Fig. 7a, b), or AURKB (Fig. 7c, d), confirmed the specificity of these antibodies and showed that PLK4 and AURKB are independently recruited to the cortex. Moreover, downregulation of either CEP192 or DVL2 inhibited recruitment of PLK4pS305 (Fig. 7a, b) and

AURKBpT232 (Fig. 7c, d) to protrusions. Taken together, these data suggest a pathway in which exosomes first promote DVL2 cortical localisation, followed by the recruitment of CEP192 to active protrusions and subsequent accumulation of active PLK4 and AURKB (Fig. 7e, f).

To further test the hypothesis that DVL2 recruits CEP192/PLK4/AURKB, we examined if we could detect CEP192, PLK4 or AURKB signal in DVL2 foci at protrusions upon ACM stimulation. For this, we generated an MDA-MB-231 cell line expressing inducible Flag-tagged DVL2 (Supplementary Fig. 7a), and then we performed fluorescence intensity correlation

**Fig. 4** Dishevelled 2 controls ACM-induced cancer cell motility by binding to PLK4, AURKB and CEP192. **a** Network graph for selected protein interactions identified from a LUMIER screen testing CEP192, PLK4 and AURKs against a collection of 3 Flag-tagged WNT-PCP and centrosomal factors. Edge width reflects the normalised LUMIER intensity ratio that indicates interaction strength (n = 2). **b** HEK293T cells were co-transfected with 3Flag-CEP192 and 3HA-PLK4 (positive control) or 3HA-DVL2 as indicated. Cell lysates were immunoprecipitated with anti-Flag antibody and 3HA-DVL2 or 3HA-PLK4 were detected by western blotting. Red arrowhead indicates band corresponding to 3HA-PLK4, blue arrowheads indicate 3HA-DVL2 bands (non-phosphorylated and phosphorylated forms), red star on the left side indicates non-specific band (n = 3). **c, d** GST, GST-AURKB (**c**) and GST-PLK4 (**d**) were bacterially expressed, purified and incubated with or without cell lysates from MDA-MB-231 cells. GST pull down was performed and endogenous DVL2 was detected by western blotting. (n = 4 for each). **e** MDA-MB-231 cells were treated with control or DVL2 siRNAs for 72 h and then stimulated with DMEM or ACM overnight. Cell motility was measured as described in Methods and analysed with one-way ANOVA Kruskal–Wallis test and post-tested with Dunn's multiple comparison test. Arrow indicates the control bar used for comparison (****p < 0.0001; n = 3, at least 40 cells were tracked per condition). **f** Cells from (**e**) were stained with actin and α-tubulin. Enlarged box areas of cortical regions are shown as ROIs in lower panels. Bar = 20 μm (n = 3). **g** MDA-MB-231 cells stably expressing tetracycline inducible C-terminally HA-tagged DVL2-wt, DVL2-N terminus (Δ361–736) or DVL2-C terminus (Δ1–360) were transfected with control or DVL2 siRNA. After 72 h, cells were treated with or without tetracycline in the presence of DMEM or ACM as indicated. Cell motility was measured as described in Methods and analysed by Mann–Whitney U test two-tailed (****p < 0.0001; n = 3, at least 50 cells were tracked per condition). All the data are plotted as box and whiskers. Boxes represent median and 25th to 75th percentiles, whiskers the minimum and maximum values with each individual cell value superimposed

profiling between DVL2 (Flag) and the signal from endogenous CEP192 or the active kinases PLK4 and AURKB at protrusions using the algorithm illustrated in Supplementary Fig. 7b (see Methods section for details). The signal correlation coefficient takes into consideration not only the overlapping area of colocalization but also the signal intensity. Our data indicate that upon ACM stimulation, the induced DVL2(Flag) signal associated with CEP192 or active PLK4 and AURKB was significantly higher than the random signal detected by the Flag antibody in uninduced cells at protrusions. This indicates that a fraction of endogenous CEP192 and the active kinases colocalize with DVL2 at the cell cortex in ACM-treated cells (Supplementary Fig. 7c–g).

We next investigated the biological significance of the interactions between components of the DVL2-CEP192-PLK4/AURKB module. As mentioned above, we observed similar binding patterns between DVL2 mutants and CEP192 (LUMIER assay, Supplementary Fig. 5c), PLK4 and AURKB (co-IP, Supplementary Fig. 5a, b; LUMER assay, Supplementary Fig. 5d, e). Moreover, depletion of DVL2 or CEP192 leads to loss of accumulation of active PLK4 and AURKB at the cell cortex in ACM-treated cells (Fig. 7a–d). We thus hypothesised that CEP192 could bridge DVL2 and PLK4/AURKB. In support of this hypothesis, we found that CEP192 depletion by RNAi significantly reduced DVL2 binding to PLK4 and AURKB using LUMIER assay (Supplementary Fig. 8a). To build on this finding, we took advantage of previous work on CEP192 isoforms, which indicated that only the longest isoform (Isoform-1, AA 1–2537) interacts with PLK4 at AA 190–240[32] (Supplementary Fig. 8k). This led us to posit that CEP192 isoform-1 specifically recruits PLK4 to cell protrusions. To test this, we first determined the binding profile of two CEP192 isoforms to DVL2, PLK4 and AURKB. Our LUMIER data show that both, CEP192 isoform-1 and a shorter isoform-2 (1–1941) interact with DVL2 and AURKB, but only isoform-1 associates with PLK4 (Supplementary Fig. 8b). Strikingly, specific knockdown of CEP192 isoform-1 (Supplementary Fig. 8c) resulted in no recruitment of active PLK4 at cell protrusions upon ACM treatment, while the recruitment of active AURKB and other CEP192 isoforms was not affected (Supplementary Fig. 8e–j). Furthermore, cells depleted of CEP192 isoform-1 treated with ACM displayed high cell motility that is dependent on AURKB, as AZD treatment inhibited cell migration (Supplementary Fig. 8d). Taken together, our data indicate that upon ACM treatment, DVL2 is required for the recruitment of all CEP192 isoforms to cell protrusions, where CEP192 isoform-1 specifically recruits active PLK4, while other CEP192 isoforms can recruit active AURKB.

Precise spatial and temporal control of AURKB is critical for its function during M phase to ensure cell cycle progression and during interphase is predominantly in the nucleus[33,34]. Since we observed active AURKB at protrusions, we hypothesised that AURKB might undergo nuclear export upon stimulation of BCCs. To test this, we measured the nuclear to cytoplasmic ratio of AURKB and observed nuclear AURKB decreased ~35% after ACM stimulation in a DVL2-dependent manner (Fig. 8a, c). Importantly, ACM stimulation did not alter cell cycle progression, indicating that the spatial distribution of AURKB in migrating BCC is a direct consequence of ACM stimulation (Supplementary Fig. 9a). Although the precise mechanism of how ACM induces DVL2-dependent AURKB nuclear export remains unclear, DVL2 is known to shuttle between the nucleus and cytoplasm[35]. DVL2 contains a conserved nuclear export sequence and a nuclear localisation sequence, and its nuclear localisation is required in the canonical WNT signalling pathway[36]. Interestingly, a recent study demonstrated that mutating DVL2 nuclear export sequence results in nuclear YAP accumulation[37]. It will be intriguing to further investigate if DVL2 also directly regulates AURKB nuclear export though a similar mechanism.

Next, we investigated the significance of AURKB nuclear export in controlling ACM-induced cell motility. AURKB is exported from the nucleus via CRM1-mediated nucleocytoplasmic shuttling[34], which can be blocked by leptomycin B (LMB). LMB treatment prior to ACM stimulation significantly inhibited AURKB nuclear export (Fig. 8b, d), but did not inhibit cell motility on its own (Fig. 8e). This was not surprising, since we established that PLK4 and AURKB act redundantly. Accordingly, we observed that the addition of both centrinone and LMB significantly reduced cell motility (Fig. 8e). These data indicate that ACM stimulation induces DVL2-dependent AURKB export from the nucleus and its subsequent accumulation at protrusions.

PLK4 levels are tightly controlled by a negative feedback loop that prevents centriole overduplication by coupling activation and autophosphorylation to degradation[38]. As noted in Supplementary Fig. 5b, co-expression of DVL2 and PLK4 significantly increased PLK4 steady-state levels, suggesting DVL2 might stabilise PLK4 at protrusions. Moreover, induced expression of DVL2 in a tetracycline-responsive cell line also noticeably increased endogenous PLK4 level (p = 0.125, Supplementary Fig. 9b, c). We thus hypothesised that DVL2 may disrupt PLK4 interaction with BTRC, the substrate-recognition moiety of the SCF complex of its E3 ubiquitin ligase[13,14]. We observed that increased expression of exogenous DVL2-wt led to increased PLK4 levels with a concomitant loss of PLK4-BTRC interaction, while expression of the DVL2 N-terminus half, that fails to bind

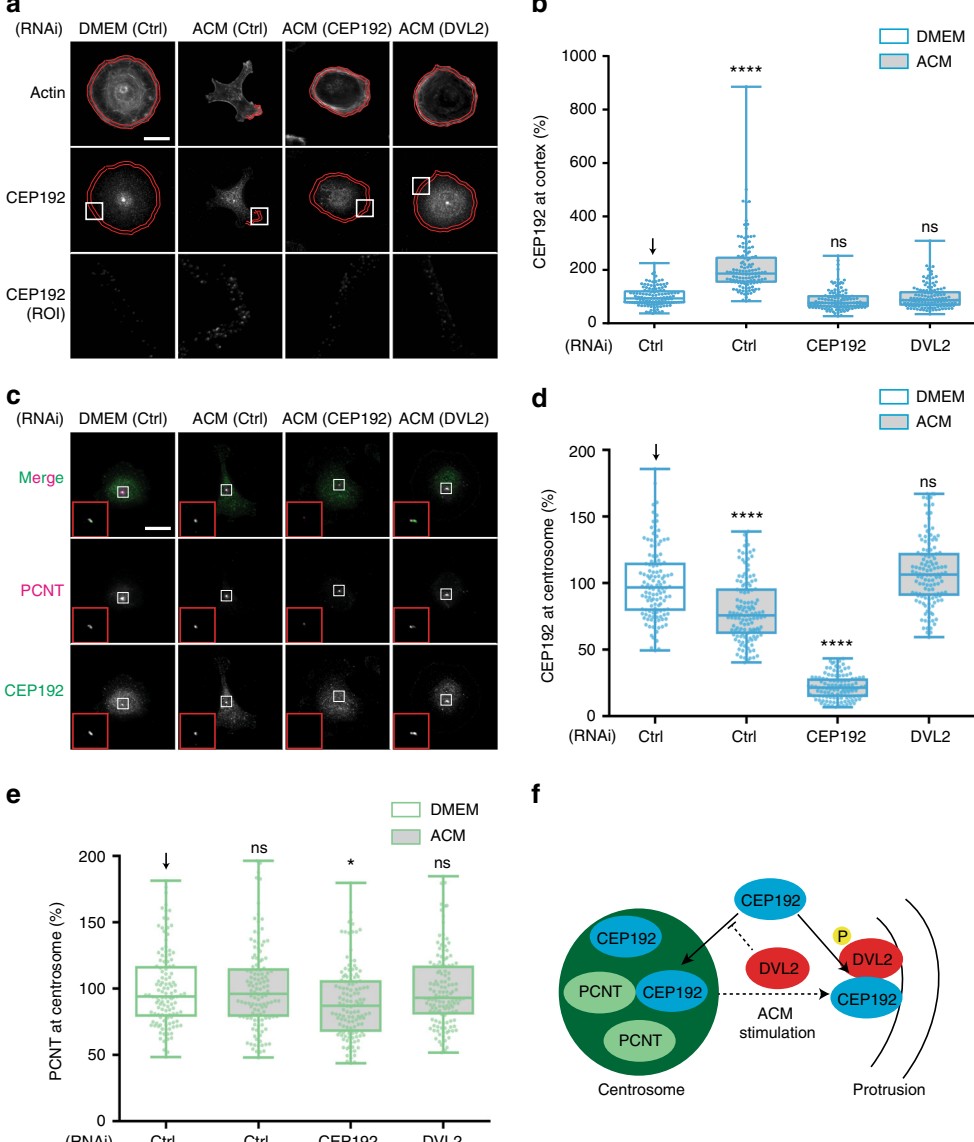

**Fig. 5** DVL2 orchestrates the recruitment of CEP192 at protrusions. **a, c** MDA-MB-231 cells were transfected with control, CEP192 or DVL2 siRNAs for 72 h, and then stimulated overnight with DMEM or ACM. Cells were stained for (**a**) CEP192 and actin, or (**c**) CEP192 and PCNT. Bar = 20 μm. **b, d, e** Intensity of CEP192 at cell cortex (**b**), at centrosome (**d**), and PCNT signal at centrosome (**e**) was measured as described in Methods and plotted as box and whiskers. Boxes represent median and 25th to 75th percentiles, whiskers the minimum and maximum values with each individual value superimposed. The data were compared with one-way ANOVA Kruskal–Wallis test and post-tested with Dunn's multiple comparison test. Arrow indicates the control bar used for comparison. (*$p < 0.05$, ****$p < 0.0001$; $n = 3$, at least 60 cells were measured per condition). Light blue is CEP192, light green is PCNT. **f** Model for regulation of centrosomal and cortical CEP192 pools by DVL2

PLK4, had no effect (Fig. 8g, h). This suggests that upon activation of the WNT signalling pathway, DVL2 recruits PLK4 to the cell cortex via CEP192 and promotes PLK4 stability at least in part by inhibiting its interaction with BTRC (Fig. 8i). Interestingly, low levels of overexpressed DVL2 increased the association between PLK4 and BTRC, although in a statistically insignificant manner (Fig. 8h comparing first and second points in blue line). This may be due to a concomitant increase of overall PLK4 levels even at low expression level of exogenous DVL2. However, we cannot exclude the possibility that other mechanisms may be involved in stabilising PLK4 under these conditions. For instance, protein phosphatases 2 A (PP2A) has been shown to stabilise PLK4[39], and directly interacts with DVL2[40]. Therefore, it is possible that the DVL2-PP2A module also contributes to PLK4 stabilisation at cell protrusions. Collectively, these studies

reveal a pathway in which DVL2 promotes AURKB nuclear export and PLK4 stability to drive assembly and recruitment of a CEP192-PLK4-AURKB module at protrusive regions during ACM induced cell motility (Supplementary Fig. 9d).

**PLK4 and AURKB regulate exchange of DAAM formins at active protrusions.** We showed that ACM-stimulated motility is an actin-dependent (Fig. 1j), but MT-independent process (Fig. 1h), that requires cortical localisation of the DVL2-CEP192-PLK4/AURKB module. We thus hypothesised that this pathway might directly link actin cytoskeletal remodelling to exosome response. The initial step of actin remodelling is actin nucleation, which is controlled via ARP2/3 complexes, formins and proteins, such as SPIRE1[7] (Supplementary Fig. 10a). We inhibited formin-

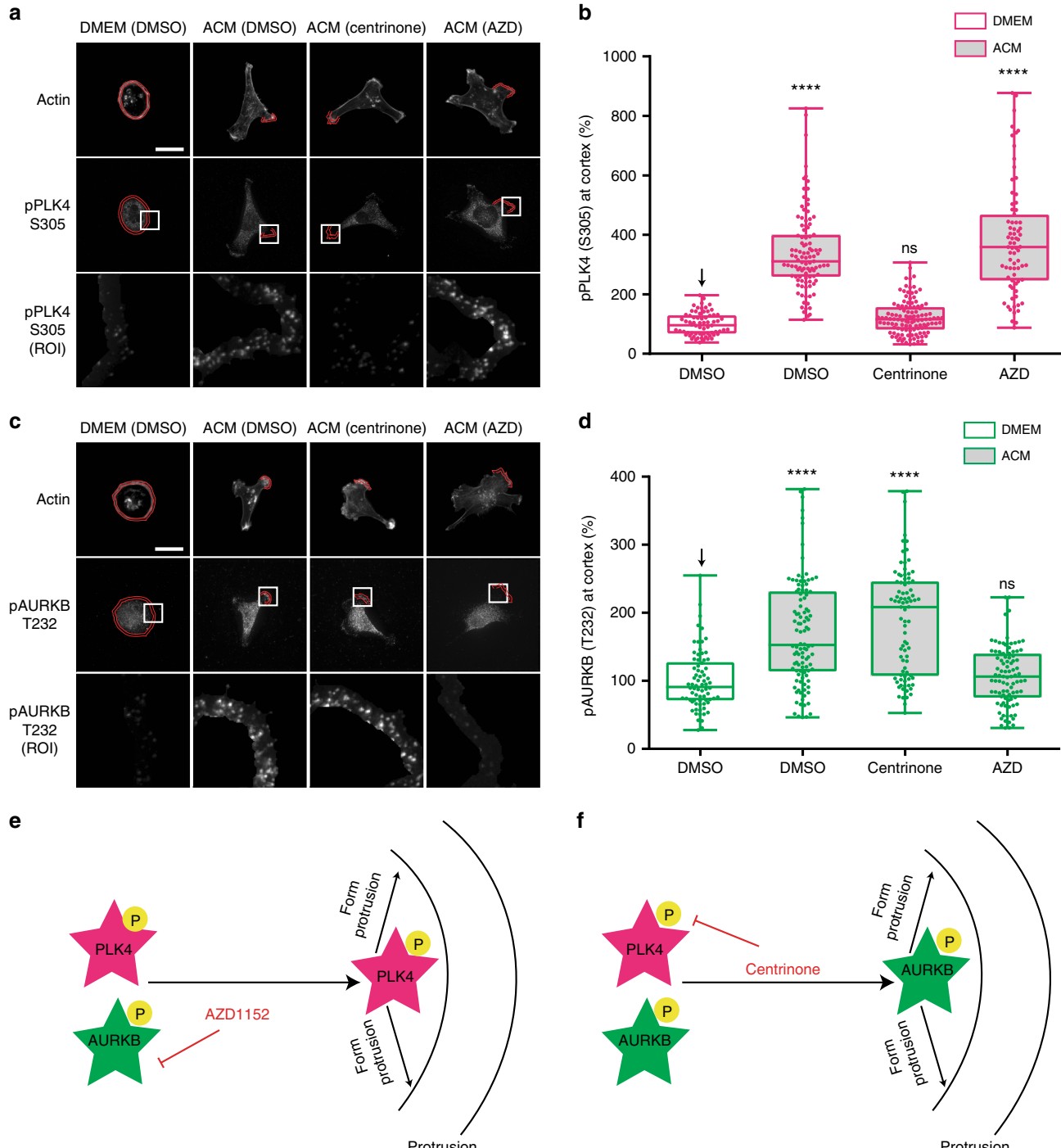

**Fig. 6** PLK4 and AURKB accumulation at cell protrusions depends on their specific activity. **a**, **c** MDA-MB-231 cells were stimulated with DMEM or ACM in the presence of DMSO, centrinone or AZD. Cells were stained for actin and (**a**) phospho-PLK4 (S305), or (**c**) phospho-AURKB (T232). Bar = 20 μm. **b**, **d** Intensity at cortical region for (**b**) phospho-PLK4 (S305), or (**d**) phospho-AURKB (T232) from cells in (**a**) and (**c**) was measured. The data was analysed as described in Methods and plotted as box and-whiskers. Boxes represent median and 25th to 75th percentiles, whiskers the minimum and maximum values with each individual value superimposed. The data were compared with one-way ANOVA Kruskal–Wallis test and post-tested with Dunn's multiple comparison test. Arrow indicates the control bar used for comparison (****$p < 0.0001$; $n = 3$, at least 60 cells were measured per condition). Magenta is phospho-PLK4, dark green is phospho-AURKB. Models for phospho-PLK4 (S305) (**e**) and phospho-AURKB (T232) (**f**) localisation at the cell cortex to promote protrusion formation that occurs independently from the other kinase activity

and ARP2/3-dependent actin nucleation using the selective inhibitors SIMFH2 or CK666, respectively (Fig. 9a), while the SPIRE1 pathway was inhibited using RNAi (Fig. 9b; Supplementary Fig. 10b). Our results showed that SMIFH2, but not CK666 or SPIRE1 RNAi, inhibited ACM-induced motility,

suggesting this is a formin-dependent process. Two formins, DAAM1 and DAAM2, have thus far been linked to DVL and WNT signalling in the context of convergent extension movements[41]. Interestingly. downregulation of DAAM1 by RNAi (Supplementary Fig. 10g) increased cell motility (Fig. 9c) in

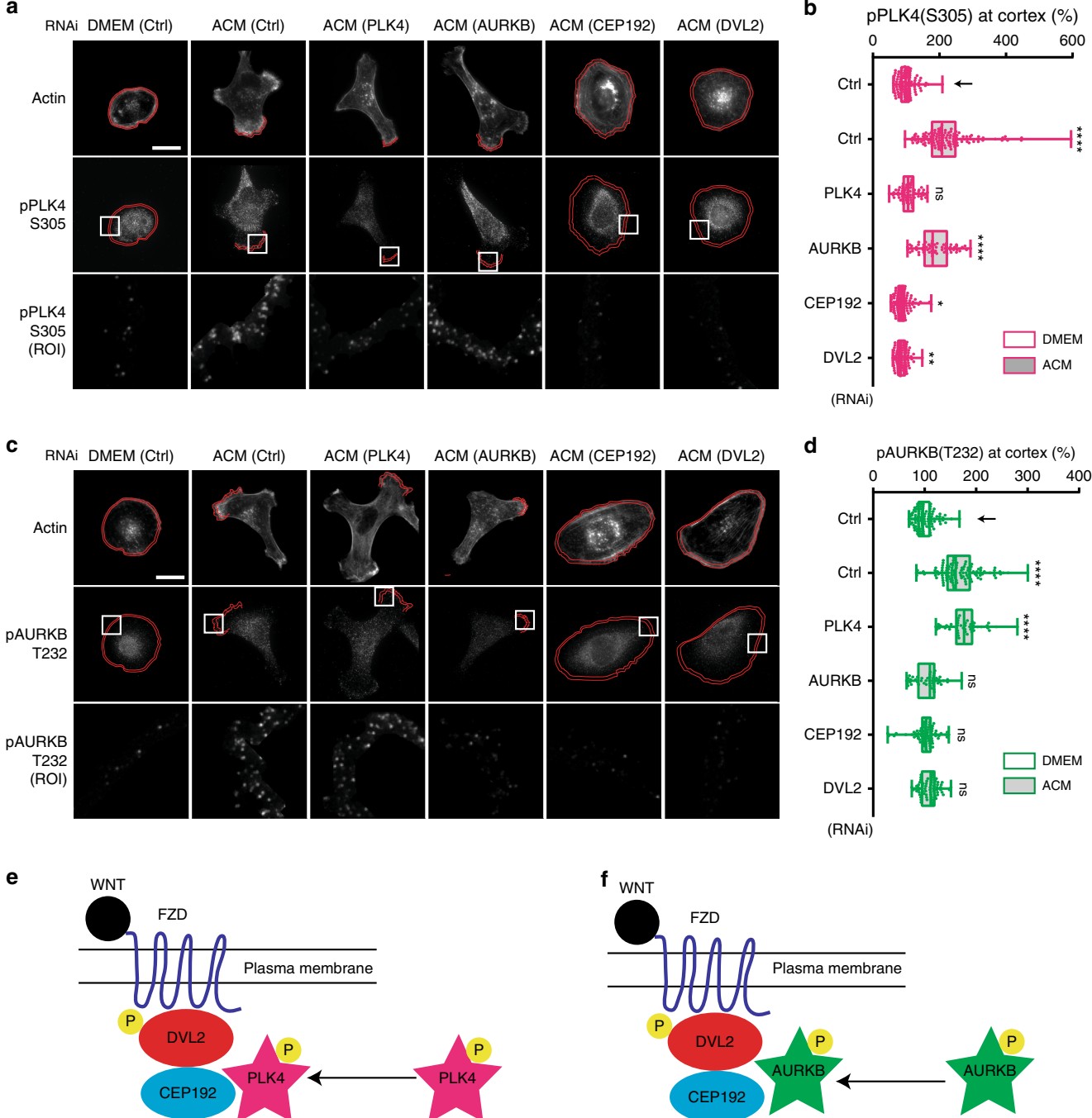

**Fig. 7** DVL2 and CEP192 orchestrate the recruitment of PLK4 and AURKB at cell protrusions. **a**, **c** MDA-MB-231 cells transfected with control, PLK4, AURKB, CEP192 or DVL2 siRNAs were stimulated overnight with DMEM or ACM. Cells were stained for actin and (**a**) phospho-PLK4 (S305), or (**c**) phospho-AURKB (T232). Bar = 20 μm. **b**, **d** Intensity of (**b**) phospho-PLK4 (S305), or (**d**) phospho-AURKB (T232) at cortical regions was measured from cells in (**a**) and (**c**), and plotted as box and whiskers. Boxes represent median and 25th to 75th percentiles, whiskers the minimum and maximum values with each individual value superimposed. The data are compared with one-way ANOVA Kruskal–Wallis test and post-tested with Dunn's multiple comparison test. Arrow indicates the control bar used for comparison (*$p < 0.05$, **$p < 0.01$, ****$p < 0.0001$; $n = 3$, at least 60 cells were measured per condition). Magenta is phospho-PLK4, dark green is phospho-AURKB. Models for (**e**) phospho-PLK4 (S305) and (**f**) phospho-AURKB (T232) recruitment and localisation at the cell cortex to promote protrusion formation via DVL2 and CEP192

unstimulated cells and was not further stimulated by ACM. In contrast, DAAM2 RNAi (Supplementary Fig. 10h) blocked ACM-induced cell motility (Fig. 9c). Similar results were obtained using three independent siRNAs targeting either DAAM1 or DAAM2. These data indicate that ACM-induced cell migration is an actin-dependent process that is regulated by the formins DAAM1 and DAAM2.

As DAAM2 was required for ACM-induced motility, and DAAM1 restrained motility in unstimulated cells (Fig. 9c), we considered that DAAM1 and DAAM2 might act in a switch-like manner during protrusion formation in motile cells. To test this, we monitored DAAM1 and DAAM2 levels, which revealed that while DAAM1 was at the cell cortex in unstimulated cells, it was depleted from ACM-induced protrusions (Fig. 9d, f). In stark

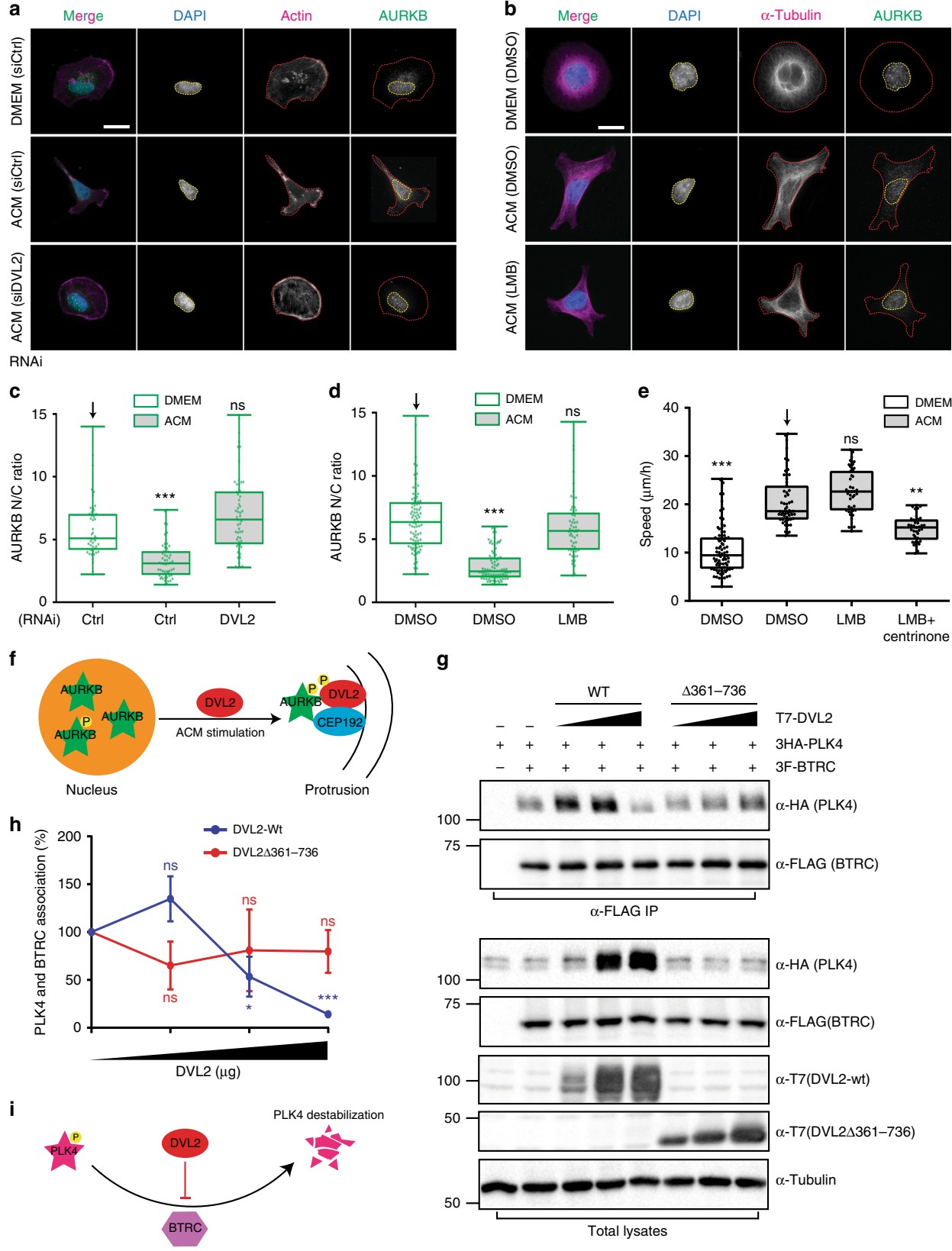

contrast, cortical DAAM2 levels were low in unstimulated cells, but markedly increased at protrusions after ACM stimulation (Fig. 9e, g). The specificity and sensitivity of the DAAM1 and DAAM2 antibodies were confirmed (Supplementary Fig. 10c–f). Furthermore, downregulation of DAAM2 prevented ACM-induced reduction of DAAM1 at the cortex (Fig. 9f), while downregulation of DAAM1 led to DAAM2 accumulation at protrusions, even in the absence of ACM stimulation (Fig. 9g) concomitant with increased basal levels of cell motility (Fig. 9c, h). We next monitored DAAM levels upon centrinone and AZD

**Fig. 8** DVL2 facilitates AURKB export from the nucleus and PLK4 stabilisation. **a** MDA-MB-231 cells were treated with control or DVL2 siRNAs for 72 h, and then stimulated overnight with DMEM or ACM. Cells were stained for actin, AURKB and nucleus (DAPI). Bar = 20 μm. **b** MDA-MB-231 cells were stimulated overnight with DMEM or ACM in the presence of DMSO or Leptomycin (LMB), a nuclear export inhibitor. Cells were stained with α-tubulin, AURKB and DAPI. Bar = 20 μm. **c, d** Quantification of the nuclear to cytoplasmic (N/C) ratio of AURKB from cells in (**a**) and (**b**), respectively. The data were plotted as described in Methods (***$p < 0.001$; $n = 3$, at least 60 cells were measured per condition). **e** Cells were stimulated overnight with DMEM or ACM in the presence of DMSO, LMB or LMB + centrinone. Cell motility was measured as described in Methods (**$p < 0.01$, ***$p < 0.001$; $n = 3$, at least 40 cells were tracked per condition). **f** Model for DVL2 regulation of the nuclear and cytoplasmic/cortical pools of AURKB. **g** HEK293T cells were transfected with 3HA-PLK4, 3 Flag-BTRC and increasing amounts of T7-DVL2-wt or T7-DVL2 (Δ361–736) mutant. Co-immunoprecipitation was performed using anti-Flag antibody. **h** Quantification of the binding between PLK4 and BTRC in the presence of increasing amounts of DVL2 is shown as average signal ratio of HA-PLK4/3F-BTRC in immunoprecipitates. Data were analysed with two-way ANOVA post-tested with Bonferroni test ($n = 4$, error bars are ± s.e.m.). **i** Model for DVL2-mediated PLK4 stabilisation. The data from **c, d** and **e** were plotted as box and whiskers. Boxes represent median and 25th to 75th percentiles, whiskers the minimum and maximum values with each individual cell value superimposed. Data were compared with one-way ANOVA Kruskal–Wallis test and post-tested with Dunn's multiple comparison test. Arrow indicates the control bar used for comparison

treatment, which showed that dual inhibition of PLK4 and AURKB sustained DAAM1 at the cell cortex in ACM-treated cells and prevented DAAM2 recruitment (Fig. 9d, e). Importantly, ACM-induced cell motility was not blocked by PLK4 and AURKB inhibition in DAAM1-depleted cells, sharply contrasting control cells (Fig. 9i and see above). These data suggest that PLK4 or AURKB activates cell migration in response to ACM stimulation, by removing DAAM1 and recruiting DAAM2 at cell protrusions. Consequently, artificial depletion of DAAM1 by RNAi can bypass the requirement of PLK4/AURKB in ACM-induced BCC migration.

Consistent with the notion that DAAM1 inhibits and DAAM2 promotes ACM-induced BCC migration (Fig. 9j), decreased expression of *DAAM1* in breast cancer patients resulted in a significant reduction in survival (Supplementary Fig. 10i). Furthermore, reduced expression of *DAAM1* and elevated expression of *DAAM2* were associated with high-grade bladder cancer (Supplementary Fig. 10j). Taken together, these data suggest a pathway in which exosomes mobilise WNT signalling at the cell cortex, which initiates a DVL2-dependent local assembly of a CEP192-PLK4/AURKB module that in turn mediates a kinase-dependent switch of DAAM1 for DAAM2 to promote protrusive activity and cell motility (Fig. 9j).

## Discussion

We have previously shown that exosome-induced BCC migration and invasive behaviour are regulated by the WNT signalling pathway which requires interplay with the PCP pathway components[9]. We also found that at the non-protrusive lateral membrane of protrusions, the PCP protein PK1 cooperates with the RhoGAPs Arhgap 21/23 to promote cell motility and protrusion formation. The work presented here defines an unexpected role for a discrete module of centrosomal proteins recruited by the WNT-PCP protein DVL2 to protrusions in response to WNT signalling, promoting exosome/ACM-driven non-directional cell migration that occurs independently of MTs and centrosomes. We find that the two major facilitators of centriole duplication (CEP192 and PLK4) and PCM assembly (CEP192) act downstream of the WNT-PCP protein DVL2 that orchestrates their recruitment to the cell cortex in response to exosome-stimulated WNT signalling. Our findings thus reveal unexpected crosstalk between these two biological pathways and define the role for this module in contextual control of cell motility.

We further show that this pathway relies on the coordinated actions of active PLK4 or AURKB kinases at the cell cortex, which are specifically recruited via different CEP192 isoforms (Supplementary Fig. 8). In addition, we demonstrate that exosome/ACM-induced cell motility is non-directional and likely its underlying mechanisms compete with directional migration (Fig. 1f, g;

Supplementary Fig. 1d, e). This is rather important, as PLK4 has been shown to regulate directional migration/invasion in either 2D and 3D cell cultures[15,17,18], by ARP2/3- and MT-dependent processes, respectively. Remarkably, we found these mediators to be dispensable in exosome/ACM-induced cell motility. The distinct roles of PLK4 in both types of migration raise the tantalising hypothesis that this kinase plays a key role in the contextual control of cell migration in response to specific stimulatory cues. Independent from PLK4 activity, AURKB is also recruited to cell protrusions by DVL2-CEP192 to activate migration. Inhibition of AURKB has been shown to suppress osteosarcoma and HepG2 cell migration and invasion[42,43], while overexpressed AURKB localises to the nucleus and cortical actin filaments of interphase normal rat kidney cells[44]. Although the emerging role of AURKB in cell migration remains to be fully defined, these studies highlight the importance of AUKRB in cell migration.

Our results suggest that at the cell cortex, AURKB and PLK4 act redundantly to regulate protrusion formation and cell migration. Consistent with this, expression of either PLK4 or AURKB in an inducible system is sufficient to promote high cell motility in the absence of ACM stimulation (data not shown). Interestingly, clinical data show that *PLK4* and *AURKB* expression are increased in high-grade bladder cancer (Supplementary Fig. 3i) and correlate with reduced survival in breast cancer patients (Supplementary Fig. 3h). Therefore, it will be important to determine whether PLK4 and AURKB also act redundantly in vivo, and if co-inhibition of PLK4 and AURKB can block or reduce metastasis, since PLK4/AURKB are predominantly viewed as cell cycle regulators and on this basis have been the subject of considerable drug development efforts. Our studies suggest that targeting contextual non-directional cell motility through dual inhibition of PLK4 and AURKB might be an important therapeutic goal in the cancer clinic.

We find that exosome-stimulated cell motility is mainly an actin-dependent process. Interestingly, an emerging, yet poorly understood property of centrosomes is their ability to act as actin-organising centre[16]. Moreover, PLK4 phosphorylates ARP2/3 to promote directional migration[17,18], while AURKB interacts with MT- and/or actin-related proteins and in turn regulates their function/localisation[45]. We therefore propose that AURKB/PLK4 may function in actin reorganisation at the cortex. Indeed, ACM-stimulated BCC migration depends on a cortical switch or exchange between DAAM1 and DAAM2, which is regulated by PLK4 or AURKB activities. Most importantly, depletion of DAAM1 by RNAi bypassed the requirement of these two kinases in cell migration, suggesting they function in the same pathway. However, how AURKB/PLK4 regulates DAAM1 removal from cell protrusions upon ACM treatment remains unknown and will be an important topic for further investigation.

Our study thus reveals an atypical function for a centrosomal module in WNT signalling and actin nucleation that is critical for

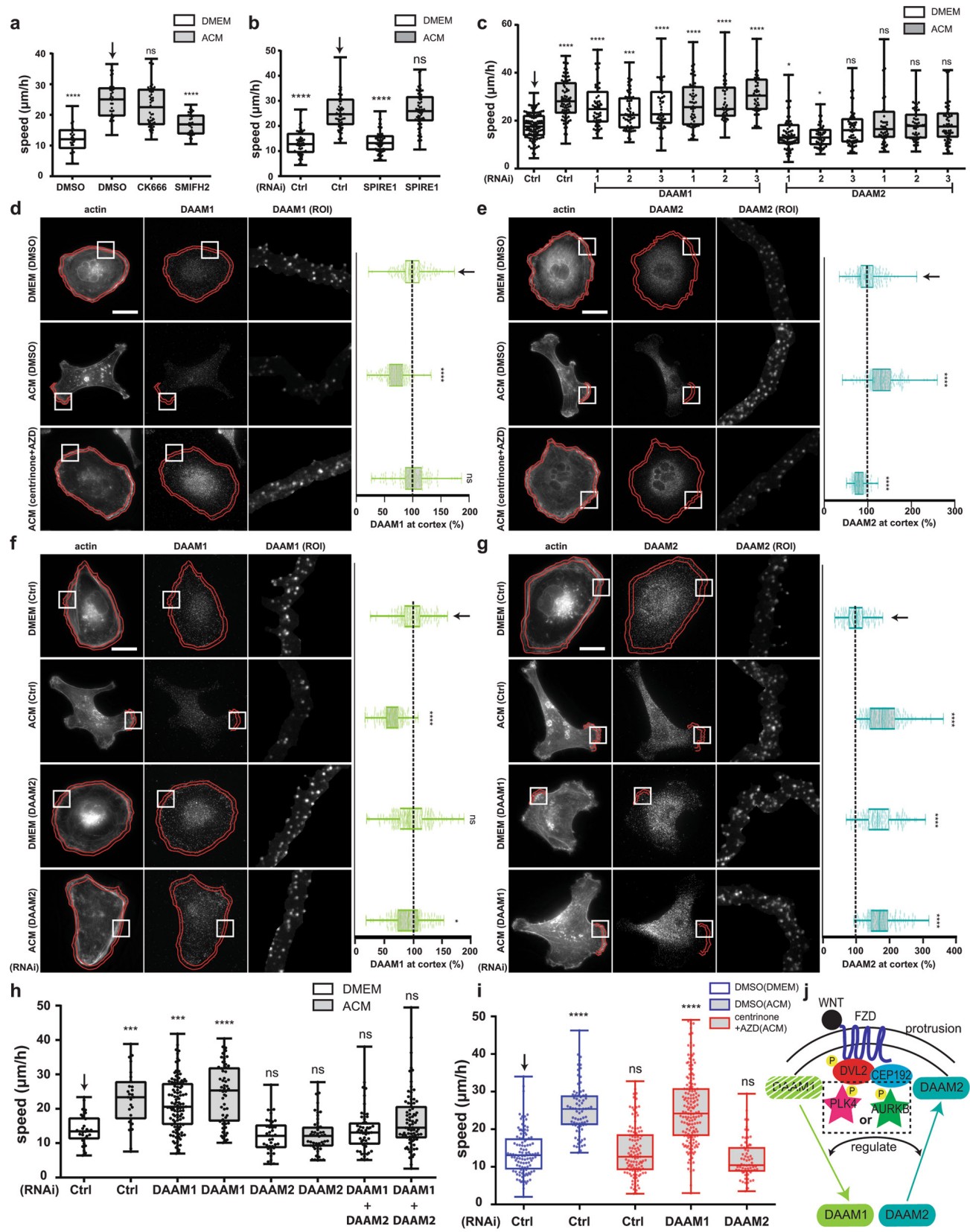

cancer cell motility. More importantly, deregulation of genes in this module including *PLK4*, *AURKB*, *DAAM1* and *DAAM2* are correlated with aggressive forms of human cancer (Supplementary Fig. 3h, i; Supplementary Fig. 10i, j). It is also notable that *DAAM1* is one of 47 genes with decreased expression in a subgroup of MDA-MB-231 cells displaying a higher propensity of lung metastasis in orthotopic models[46]. Moreover, topographic single cell sequencing comparison between in situ versus invasive ductal carcinoma breast cancers revealed that *CEP192* is one of eleven misregulated genes of an invasive signature from a specific

**Fig. 9** PLK4 and AURKB regulate cell motility through DAAMs. **a** MDA-MB-231 cells were stimulated overnight with DMEM or ACM in the presence of DMSO, CK666 (ARP2/3 inhibitor) or SMIFH2 (formin inhibitor). **b** Cells were transfected with control or SPIRE1 (actin nucleator) siRNAs for 72 h and then treated overnight with DMEM or ACM. Cell motility from cells in (**a**) and (**b**) was measured as described (****$p < 0.0001$; $n = 3$, at least 50 cells were tracked per condition). **c** Cells were transfected with control, DAAM1 or DAAM2 individual siRNA oligos for 72 h and then stimulated overnight with DMEM or ACM as indicated. Cell motility was measured as described in Methods (*$p < 0.05$, ***$p < 0.001$, ****$p < 0.0001$; $n = 3$, at least 50 cells were tracked per condition). **d, e** MDA-MB-231 cells were stimulated with DMEM or ACM in the presence of DMSO or centrinone + AZD as indicated. Cells were stained for actin and (**d**) DAAM1, or (**e**) DAAM2. Bar = 20 μm. Right side box and whiskers plots show the quantification of average signal intensity at the cell cortex, which was measured as described in Methods. Light green is DAAM1, teal is DAAM2 (****$p < 0.0001$; $n = 3$, at least 60 cells were measured per condition). **f, g** Cells were transfected with control, DAAM2 or DAAM1 siRNAs for 72 h, and then stimulated overnight with DMEM or ACM as indicated. Cells were stained for actin and (**f**) DAAM1, or (**g**) DAAM2. Bar = 20 μm. Right side plots show the average signal intensity at cortical region for DAAM1 (light green) or DAAM2 (teal), which was measured and plotted as described in Methods (*$p < 0.05$, ****$p < 0.0001$; $n = 3$, at least 60 cells were measured per condition). **h** MDA-MB-231 cells were transfected with control, DAAM1, DAAM2 siRNAs or the indicated combinations for 72 h and then stimulated overnight with DMEM or ACM. Cell motility was measured as described (***$p < 0.001$, ****$p < 0.0001$; $n = 3$, at least 60 cells were measured per condition). **i** MDA-MB-231 cells were transfected with control, DAAM1 or DAAM2 siRNAs for 72 h and then stimulated overnight with DMEM or ACM in the presence of DMSO or centrinone + AZD. Cell motility was measured as described (***$p < 0.001$; $n = 3$, at least 60 cells were tracked per condition). **j** Model for DAAM1 and DAAM2 localisation switch at cell cortex upon ACM treatment regulated by PLK4 and AURKB. All the data were plotted as box and whiskers. Boxes represent median and 25th to 75th percentiles, whiskers the minimum and maximum values with each individual cell value superimposed. The data were compared with one-way ANOVA Kruskal–Wallis test and post-tested with Dunn's multiple comparison test. Arrow indicates the control bar used for comparison

mutation[47], further supporting the role of this module in cancer migration and invasion.

Taken together, we provide evidence of a centrosomal module that performs functions in the WNT signalling pathway that are independent from its centrosomal roles, by directly regulating actin reorganisation via formins and in turn promoting cancer cell migration. It is tantalising to speculate that this pathway may, at least in part, regulate other functions unrelated to cancer. For example, DVL has been linked to basal body docking and orientation in ciliated epithelial cells[48]. Considering that *PLK4* and *DAAM2* are overexpressed 20- and 3-fold, respectively, during multiciliation[49] and that PCP proteins have been implicated in the control of ciliary beating and ciliogenesis[50,51], it is intriguing to consider that the centrosomal module identified here may function in developmentally important cellular processes, such as ciliogenesis. Furthermore, disruption of PLK4 function leads to defects in frog somite morphogenesis and eye development implying a role in the PCP pathway[52]. Although AURKB−/− embryos are normally implanted, consistent with our findings, AURKC replaces AUKRB function during embryogenesis[53], and therefore the importance of AURKB in embryonic development may still be under appreciated. Indeed, in zebrafish that lack the AURKC ortholog, AURKB localises within motor neurons in the developing spinal cord and regulates axonal outgrowth and guidance[54], which is known to be mediated by the WNT-PCP pathway. Therefore, it will be intriguing to further investigate the role of AURKB and PLK4 in human embryo/tissue development, especially if there are redundant functions of these kinases that remain to be uncovered. Finally, whether the centrosomal module identified here regulating contextual control of cell motility has any roles in other developmental and myriad other disease-related contexts will be an important area for further research.

## Methods

**Cell culture**. MDA-MB-231 cells were a gift from Dr. Robert S. Kerbel (Sunnybrook Health Sciences Centre, Toronto, Canada)[9] and cultured in RPMI (GIBCO) supplied with 5% foetal bovine serum (FBS) (Gibco). L cells were purchased from ATCC (CRL-2648™) and cultured in Dulbecco's modified Eagle's medium (DMEM) (HyClone) with 10% FBS. HEK293T cells were purchased from ATCC (CRL-3216™) were cultured in DMEM with 10% FBS supplemented with GlutaMAX. MDA-MB-468 cells were purchased from ATCC (HTB-132™) and were cultured in the DMEM/F-12 (1:1) medium (Hyclone) with 10% FBS. Mouse breast cancer cell lines EpRas were a gift from Dr. Martin Oft and were cultured in the DMEM (HyClone) with 10% FBS. Human bladder cancer cell lines T24 (HTB-4™) and TCCSUP (HTB-5™) were purchased from ATCC and cultured in

McCoy 5 A (Thermo Fisher) and RMPI (GIBCO) with 5% FBS, respectively. All cells were maintained at 37 ºC in a humidified atmosphere containing 5% $CO_2$. During the course of this study, all cell lines were routinely tested for mycoplasma contamination. All cell lines were authenticated by STR profiling.

**Active-conditioned media (ACM) preparation**. Confluent L cells were incubated with serum-free DMEM supplemented with penicillin/streptomycin and amphotericin B. Three days later, the supernatant (ACM) was collected, filtered and stored at 4 ºC for up to 2 months[9].

**Exosome isolation and stimulation**. Exosomes were isolated by filtering the ACM, then using sequential ultracentrifugation 10,000 × g for 40 min, then 100,000 × g for 4 h, resuspended with PBS, and purified by a final centrifugation at 100,000 × g for 2 h. The amount of purified exosomes used for stimulation is equivalent to what was used in motility assays done with ACM.

**Cell motility assay**. Cells were seeded at a very low density in assay media prepared by mixing equal volumes of RPMI (5% FBS), this is at a 1:1 ratio, with DMEM or ACM[9]. Phase-contrast time-lapse movies were collected on a DeltaVision (DV) Elite high-resolution Microscope (Applied Precision) using a ×20 N-Plan objective lens (Olympus). Cells were maintained at 37 ºC in a humidified atmosphere containing 5% $CO_2$. Cell speed (in μm/h) was measured by manually tracing the centre of the cell nucleus every 10 min for 16 h using ImageJ. The average speed of cells during the total imaging time was plotted as box and whiskers. The box extends from the 25th to 75th percentiles, the middle line of the box is plotted at the median, while the whiskers show the minimum and maximum values with each individual value plotted as a dot superimosed on the graph. The data were compared with one-way ANOVA Kruskal–Wallis test and post-tested with Dunn's multiple comparison test. For running average speed plots in Figs. 1h, j and 3e, a cell was continuously tracked before and after drug treatment. The running average of cell speed for every hour is plotted. The average of at least 40 cells per treatment is presented ( ± s.e.m.).

**Cloning and transfection**. Human CEP192 coding sequences (NM_032142 long isoform-1 and Q8TEP8–1 short isoform-2) were PCR amplified and recombined into pDONR223 vector using Gateway system (Invitrogen/Thermo Fisher). Human PLK4, AURKA, AURKB and AURKC entry vectors were obtained from OpenFreezer[55]. CEP192, PLK4, AURKB and AURKC were tagged at the N-terminus with Renilla Luciferase (RL), 3Flag or 3HA by recombination using the Gateway system. Human 3Flag-DVL2 wild-type and deletion mutants were generated by directed mutagenesis[56,57]. Poor expression of smaller fragments of DVL2 precluded a more detailed analysis of deletion mutants. Cells were transfected with cDNAs using PolyFect (Qiagen) or Lipofectamine LTX with Plus reagent (Invitrogen) according to the manufacturer's protocol. RNAi was transfected at a final concentration of 20 nM using Lipofectamine RNAiMax (Invitrogen) according to the manufacturer's protocol.

**Generation of stable MDA-MB-231 cell lines**. Lentiviral transfer vectors pSIN-GFP-CETN2 were created by cloning CETN2 into pSIN-GFP vector. DVL2 wild-type, Δ361–736 and Δ1–360 fragments were cloned into pInducer20, a Tet-inducible lentiviral vector[58]. HEK293T cells were transfected with lentiviral transfer vectors in addition to psPAX2 and pVSVG using Lipofectamin 3000 (Invitrogen). Twenty four hours post transfection, media was replaced with high-serum (20% FBS) media and

virus was collected after 48 h. MDA-MB-231 cells were then infected with viral supernatant. GFP-CETN2-positive cells were sorted by FACS, and DVL2 cell lines were selected by G418.

**LUMIER protein interaction screening**. We performed LUMIER screens[27] to discover interactors using as baits Renilla luciferase (RL) fusions for CEP192, PLK4, AURKB and AURKC. They were tested against 79 Flag-tagged WNT-PCP pathway components, and 5 known interactors of the baits (positive controls). HEK293T cells were co-transfected in 96-well plates with a unique bait and Flag-tagged prey combination per well. After 48 h, cell lysates were immunoprecipitated in plates coated with M2 anti-Flag antibody (Sigma cat# F-1804). Renilla luciferase substrate was added to detect the presence of the bait in the immunoprecipitates. The screens were performed in duplicates (biological replicates), and positive interactions with NLIR values ≥ 6[59,60] in both screens were selected for confirmation by co-immunoprecipitation and western blot, using 3HA-tagged baits expressed in HEK293T cells.

**Immunoprecipitation and western blotting**. HEK293T cells were transfected using PolyFect (QIAGEN), and 48 h later they were lysed using TNTE 0.5% buffer (50 mM Tris-HCl pH 7.4, 150 mM NaCl, 1 mM tetrasodium EDTA, 0.5% Triton X-100) in the presence of protease inhibitors. Cleared supernatants were incubated with M2 anti-flag antibody (Sigma) for 1–2 h rocking at 4 °C. Protein G-Sepharose (GE Healthcare) was added, and then samples were rocked for 1 h with the beads. Protein immunecomplexes were washed using TNTE 0.1% buffer (50 mM Tris-HCl pH 7.4, 150 mM NaCl, 1 mM EDTA, 0.1% Triton X-100). For western blotting, protein lysates in Laemmli buffer were loaded on 7.5%, 8% or 12% SDS polyacrylamide gels and then transferred to 0.45- μm nitrocellulose membranes (Biorad, cat#1620115) or PVDF (Biorad, cat#1620177). After transfer, the membranes were blocked with 5% non-fat dry milk in Tris buffered saline containing 0.1% Tween (TBST) and incubated with primary antibodies overnight at 4 °C. Membranes were incubated with goat anti-rabbit IgG coupled with horseradish peroxidase (1:10,000, GE Healthcare cat# RPN4301) or sheep anti-mouse IgG coupled with horseradish peroxidase (1:10,000, GE Healthcare cat# RPN4301) and developed with Femto (Thermo Scientific) or Clarity™ Western ECL Substrate (Biorad). Images were acquired and quantified where indicated with Biorad Chemidoc™ MP Imaging System. For publication purposes, original files were exported as 300 dpi TIFF images, sporadically brightness and contrast settings were adjusted for the whole image without disturbing the image integrity (no Gamma settings were changed). When HEK293T or MDA-MB-231 cells were transfected with RNAiMax (Life Technologies/Thermo Scientific), protein concentration was measured using the Pierce BCA Protein Assay Kit (Thermo Scientific) to load equal amounts per lane for SDS-PAGE. Uncropped and unprocessed scans of the blots from main figures can be found in Supplementary Fig. 11.

**Microscopy**. Super-resolution microscopy was performed on a three-dimensional structured-illumination microscope (3D-SIM) (OMX Blaze v4, GE Biosciences PA) as described[61]. Multi-channel 3D-SIM image Z stacks were reconstructed, three-dimensionally aligned using calibrations based on a GE reference slide and 100 -nm diameter TetraSpeck Microspheres and maximum intensity projected using the softWoRx 6.0 software package (GE).

**Immunofluorescence**. Cells were treated with siRNA or kinase inhibitors as indicated, and then seeded on coverslips and supplemented with the DMEM or ACM for overnight treatment. Cells were fixed with 4% PFA in the presence or absence of 0.1% glutaraldehyde, then permeabilized in 0.2% Triton/PBS with or without a prior quenching step for 7 min with 0.1% NaBH₄ (Sigma), respectively, and blocked in 0.2% gelatin-2%BSA/PBS. Samples were incubated with primary antibody for 2 h and with secondary antibody for 45 min at room temperature. Samples were washed with 0.1% PBS-Tween 20 between each incubation. For all quantitatively compared images, identical exposure times and imaging conditions were maintained, and maximum/average intensity projections of Z stacks were used as indicated in Quantification of immunofluorescence images.

**Quantification of immunofluorescence images**. To quantify the intensity of protein of interest (POI), 30 z-planes of 6 μm from more than 50 cells per sample were acquired at 60 ×/1.4NA on a Deltavision Elite DV with a 2048 × 2048 sCMOS camera (GE Life Sciences, PA). Image analysis employing adaptive thresholding was carried out on deconvolved, z-projected stacks with MATLAB to identify protrusions, centrosome, nucleus or cytoplasm using specific markers as described below. The data were presented as box and whiskers as described in Cell Motility Assay. The data were compared using Kruskal–Wallis test and post-tested with Dunn's multiple comparison test.

For quantification of the cell aspect ratio, actin and nucleus (DAPI) signal (average projecrion) was used to segment cells using MATLAB (MathWorks). The aspect ratio of each cell was then measured.

For quantifications of the cell cortex and protrusions, and as the diagram in Supplementary Fig. 6b shows, the cell outline was segmented according to the actin signal (average projection) using MATLAB (MathWorks). Then a ~2-μm-thick ring was generated by walking from cell outline towards the cell centre to isolate

the cortical region. This cortical region mask generated from the actin signal was then applied to protein of interest (POI). For the cells with protrusions, the protrusions were manually sliced off in ImageJ using actin signal as a reference. For the cells that failed to make protrusions, we first dilated the nucleus using DAPI to remove the cortical regions that were close to the nucleus so that selected regions would be similarly distanced from the nucleus as the regions selected for cells with protrusions. After removing these areas, the algorithm randomly picks two boxes per cell, which have the same size as the ones used in selecting protrusions, from the remaining cortical region. The mean signal intensity of POI (maximum projection) under the selected region was then measured.

For the analysis of colocalization between DVL2–3Flag and endogenous CEP192, PLK4 or AURKB, signal quantification was performed in MDA-MB-231 cells expressing Tet-inducible DVL2–3Flag. Cells were treated with tetracycline (0.5 μg/ml) and ACM overnight and processed for immunofluorescence using a mouse anti-flag antibody and the specific rabbit antibodies recognising CEP192, active PLK4 and active AURKB to detect all the POIs at cell protrusions. Secondary goat anti-mouse AlexaFluor® 594 (Channel 1) and goat anti-rabbit AlexaFluor® 647 or goat anti-mouse AlexaFluor® 647 (Channel 2) was then applied. We designed an algorithm illustrated in Supplementary Fig. 7b in which, after manually cropping the protrusions, masks were generated to capture the signals for Flag (DVL2) (Channel 1) and the other POI (Channel 2). The masks were then combined and the correlation coefficient of the two channels was calculated. As the normalising reference, we used the correlation coefficient of the induced DVL2(Flag) signal colocalizing with itself at cell protrusions in both channels using two non-overlapping fluorescently labelled secondary antibodies (anti-mouse AlexaFluor® 594, and anti-mouse AlexaFluor® 647) recognising the Flag (DVL2) signal.

For centrosomal quantification, the centrosome mask was generated using the pericentrin (PCNT) signal (maximum projection) in MATLAB. The mask of the centrosome was then applied to the POI (maximum projection) and the total intensity was measured.

For quantification of the nucleus and cytoplasm, the cell nucleus was selected using the signal from DAPI staining (average projection) and the whole cell was segmented using either actin or tubulin signal (average projection) in MATLAB. The mean intensity of POI (maximum projection) at the nucleus or cytoplasm (defined as the whole cell excluding nucleus region) was then measured.

To avoid fluorescence signal varations from different experiments, all the images from a given experiment were collected the same day using identical imaging parameters, and the POI intensity was normalised to control cells from the same experiment as percentage (fold) changes.

**Wound-healing assay**. MDA-MB-231 cells were treated with inhibitors or siRNA as indicated. The day before imaging, 840,000 cells were seeded into six-well plates with 0.5% serum. DMEM or ACM was added to each well in a 1:1 ratio, if needed. A single wound was made the next day with a P20 pipette tip, and then cells were transferred to the DV microscope as described above. Live imaging was captured using a ×10 objective lens (Olympus) at every 30 min up to 24 h, and 10 randomly picked locations of a single wound were imaged. Image analysis was carried out by ImageJ measuring the wound area at t = 0, 6, 12, 18 and 24 h. The residual wound area was expressed as a percent of original wound at t = 0.

**Matrigel 3D cultures**. MDA-MB-231 cells were harvested 48 h after RNAi transfection using trypsin and transferred as a single cell suspension culture media which was then mixed with growth factor reduced Matrigel (BD BioSciences) Matrigel to achieve a final 75%v/v Matrigel. Cells were then incubated with assay media for further 24 h before imaging. Live imaging was then captured as described in Cell motility assay section, at every 15 min during a 16 h incubation.

**In vitro protein interaction assay**. Human PLK4 was obtained from Addgene pFastBac GST-PLK4 (plasmid # 80264[62]), and human AURKB was amplified from an entry vector obtained from Openfreezer[55]. Both PLK4 and AURKB full-length were subcloned into BamHI-NotI restriction enzyme sites in bacterial expression vector pET30M and sequenced-verified. PLK4 and AURKB were expressed as a GST-fusion, the fusion protein expression was induced with 0.5 mM IPTG overnight at 17 °C, bacterial culture harvested and lysed in a buffer containing 50 mM Tris-HCl pH 7.5, 150 mM NaCl, 1 mM DTT and cocktail of protease inhibitors. GST-PLK4 or GST-AURKB fusion protein was pulled down with glutathione Sepharose beads (GE Healthcare). GST-PLK4 or GST-AURKB pulling down experiments for endogenous DVL2 from MDA-MB-231 lysates were performed in TNTE buffer containing 50 mM Tris-HCl (pH 7.5), 150 mM NaCl, 1 mM EDTA, 1 mM PMSF and 0.1% Triton X-100. The GST-fusion proteins were incubated with the MDA-MB-231 lysates for 4 h at 4 °C. Samples were washed with TNTE and processed for western blotting[56].

**Real-time quantitative PCR**. To confirm the knockdown of proteins of interest when no suitable antibodies could be found, we performed real-time-quantitative PCR (RT-qPCR)[10]. The total RNA was extracted from MDA-MB-231 cells using the RNeasy Mini Kit (QIAGEN) and reveres transcribed using oligo (dT) primers and SuperScript III Reverse Transcriptase (Life Technologies). Real-time PCR was conducted using SYBR Green master mix (Roche) in a Light Cycler 480 (Roche).

Gene expression levels were normalised to human HPRT and quantified using the $2^{-\Delta\Delta CT}$ method. The primers used were as follows: hAURKC[63]; forward: 5′- TATAACTATTTCCATGATGCACGCC-3′, reverse: 5′-ACTTTCTTGTCATG GCAGTAGGTC-3′. hHPRT[64]; forward: 5′-TCCAAAGATGGTCAAGGTCGCAA G-3′ reverse: 5′- TGGCGATGTCAATAGGACTCCAGA-3′.

**Flow cytometry**. MDA-MB-231 cells were cultured with DMEM or ACM over-night. Approximately $1–2 \times 10^6$ cells were then harvested as a single-cell suspension, washed twice in ice-cold PBS and re-suspended in 1 -ml ice-cold PBS in a 15-ml v-bottom tube (Falcon 352096). In total, 3 ml of cold ethanol ($-20\ ^oC$) was then added dropwise while vortexing to minimise clumping and the cells fixed at $-20\ ^oC$ overnight. Fixed cells were then washed twice in PBS and re-suspended in a 1 -ml solution of PI (40 µg/ml) and RNAse A (50 µl of 10 mg/ml stock) and incubated for a minimum of 3 h at 4 ºC. Cells were then acquired on a flow cytometer (Beckman Coulter Gallios 10 colour/4laser) to measure DNA content (PI intensity). DNA content was modelled using ModFit LT V4.1 (Verity Software House) with gates to remove acquisition irregularities (time vs. PI lin) and doublets (PI Integral lin. vs. PI Peak lin.).

**Bladder cancer RNA-seq**. The study was carried out with institutional ethics board approval from two institutions (University Health Network: 11–0134-T and Mount Sinai Hospital: 11–0015-E). Formalin-fixed paraffin-embedded (FFPE) archival urothelial bladder cancer (UBC) specimens from the University Health Network (UHN) departmental database were subjected to serial sectioning. The FFPE sections were reviewed by a pathologist to confirm the assigned pathological grades. The total RNA was extracted from the FFPE tumour samples; and cDNA libraries were prepared and sequenced by HiSeq2000. Transcript reads were mapped onto the human genome (hg19), and expression levels of each gene were estimated by counting the number of reads mapped to the gene normalised to total mapped reads to obtain transcript union Reads Per Million total reads (truRPMs)[65].

**Antibodies, siRNAs and other reagents**. Phospho-PLK4 (S305) antibody (1:500) was a gift from Dr. Tak W. Mak (Princess Margaret Cancer Centre). Phospho-AURKB (T232) antibody (1:200, cat# 636101) was purchased from BioLegend. CEP192 antibody (1:500 Immunofluorescence) was generated as described[24]. Phospho-DVL2 (S143) (1:500, cat# ab124933), PCNT (1:1000, cat# ab4448), and AURKA (1:1000, cat# ab12875) antibodies were purchased from Abcam. Anti-CEP192 antibody (1:500 Western blotting, cat# A302–324A) was purchased from Bethyl Laboratories. Anti-AURKB (1:500, cat# 611082) antibody was purchased from BD Biosciences. Anti-PLK4 (1:500, cat# MABC544) was purchased from Millipore. Anti-alpha tubulin (1:500 Immunofluorescence, 1:10000 western blotting, cat# T9026), anti-gamma tubulin (1:500, cat# T6557), anti-beta actin (1:1000, cat# A5316), anti-flag (1:10000, cat# F1804), anti-GST (1:5000, cat# G7781) and anti-HA-HRP (1:10000, cat# 11867423001) were purchased from Sigma. DAAM1 (1:500, cat# sc-100942) was purchased from Santa Cruz. Antibody for DVL2 (1:500 Immunofluorescence, 1:1000 Western blotting, cat# 3324) was purchased from Cell Signalling Technology. DAAM2 (1:500, cat# NBP2–47496) and SPIRE1 (1:500, cat# H00056907-M01) antibodies were purchased from Novus Biologicals. Alex-aFluor™ 488 Phalloidin (A12379) was purchased from Thermo Fisher. Additionally, polyclonal phospho-PLK4 (S305) (1:5000) and phospho-PLK4 (T170) (1:1000) antibodies were generated by Covance. Peptides (SISGpSLFDKRRLLC and HEKHYpTLCGTP, for PLK4 S305 and T170, respectively) were conjugated to KLH and used to immunise rabbits.

The siRNA control was the GL2 Duplex targeting Luciferase (CGUACGCGGAAUACTTCG) from Dharmacon. The siRNAs targeting human PLK4 (L-005036–00), CEP192 (all isoforms) (L-032250–01), PRICKLE (L-016677–02), NEDD1 (L-008306–01), AURKA (L-003545–01), SPIRE1 (L-023397–02) and were SMART pools from Dharmacon (Supplementary Table 2). The siRNAs targeting human AURKB (CCAAACUGCUCAGGCAUAA) and AURKC (GGGAGAACAUAUGAUGAAA) were ON-TARGETplus from Dharmacon. DVL2 was depleted using STEALTH siRNA targeting the 3′untranslated region CACAGUGGCCACAAUCUCCUGUAUG from Life Technologies. The siRNA targeting CEP192 isoform-1 N-terminal region (5′-GCUUAAACUGCAAGUUUCAAUCAGA-3′) was from Life Technoloies[22,66]. The siRNAs targeting DAAM1 and DAAM2 were ON-TARGETplus from Dharmacon, with the following sequences: GGACCGAAAUUGAUGAUAC (DAAM1–1), GUGGAGAGUUUGGGACUUUA (DAAM1–2), AGAUAGGCAU UUAGACUUU (DAAM1–3), GCAAUAAACCGGCUAAAUU (DAAM2–1), GGCUUGACCUGUCUGCUAA (DAAM2–2) and GGACACAGCCUAUGAGGUU (DAAM2–3).

PLK4 kinase inhibitors centrinone and centrinone B[19] were gifts from Dr. Andrew Shiau (Ludwig Cancer Research). The PLK4 inhibitor CFI-400945 and AURKB/C kinase inhibitor AZD1152[67] were gifts from Dr. Tak W. Mak (Princess Margaret Cancer Centre). Tetracycline was purchased from Bioshop. Leptomycin B (L2913), Cytochalasin B (C6762), Cytochalasin D (C8273), CK-666 (SML0006) and SMIFH2 (S4826) were purchased from Sigma-Aldrich. Propidium Iodide (P4170) was purchased from Sigma. RNase A (LS005649) was purchased from Worthington Biochemicals.

**Statistics and reproducibility**. All quantitative data are presented as box-and-whisker plots, with the boxes extending from the 25th to 75th percentiles, and the line in the middle of the box indicating the median. The whiskers go from the minimum up to the maximum values and all individual data points analysed per condition are superimposed on each plotted bar. Statistical tests were performed using GraphPad Prism 7.04. Mann–Whitney U test two-tailed were used for comparisons of the means of data between two groups. For multiple independent groups, one-way ANOVA Kruskal–Wallis test subsequently post-tested with Dunn's multiple comparison test was used. Values of $P < 0.05$ were considered significant. Sample size was generally chosen on the basis of preliminary data indicating the variance within each group and the differences between groups.

**Reporting summary**. Further information on research design is available in the Nature Research Reporting Summary linked to this article.

## Data availability

The RNA-seq data generated and analysed from bladder cancer FFPE samples are available at the NCBI GEO database with the accession number GSE129871. The data from breast cancer patients were obtained from the TCGA website at http://www.cbioportal.org[68,69]. All other data generated during this study are available from the corresponding author upon reasonable request.

## Code availability

The algorithms generated for the quantification of proteins of interest at the cell cortex were generated using MATLAB version r2015b with the image analysis toolbox. The code is available from the corresponding author upon reasonable request.

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

## Acknowledgements

We thank Dr. T.W. Mak, and Jacqueline Mason for providing the phospho-PLK4 (S305) antibody, PLK4 inhibitor CFI-400945, AURKB inhibitor AZD1152 and helpful discussions. We thank Dr. D. Schramek and Geraldine Mbamalu for providing MCF10A cells and M. Parsons at the LTRI Flow Cytometry Facility for training, technical support and help in generating FACS data. Dr. Alessandro Datti and Jenny Wang at the SMART facility (LTRI) for assistance with LUMIER screening. Jess Shen and Kin Chan at the Network Biology Collaborative Centre (https://nbcc.lunenfeld.ca) for the bladder cancer RNAseq data. Dr. K. Oegema for initially providing the PLK4 centrinone inhibitors. We thank Dr. C.J. Swallow for discussing unpublished work and providing the ARP2/3 inhibitor CK-666. 3D-SIM and time-lapse imaging were performed at the Network Biology Collaborative Centre (NBCC), a facility supported by Canada Foundation for Innovation, the Ontarian Government, and Genome Canada and Ontario Genomics Institute (OGI-139). We thank Dr. Liliana Attisano and Sally Cheung for comments on the manuscript and Pelletier and Wrana laboratory members for helpful suggestions. This work was supported by the CCSRI (Innovation grant 704378) to L.P., the Krembil Foundation and the CFREF-MbD to L.P. and J.L.W., the CCSRI (Impact grant 702320) to L.P. and a CIHR Foundation grant (FDN 143252) to J.L.W.

## Author contributions

Y.L. designed research, performed experimental work, and wrote the paper. M.B.R. performed LUMIER screen and co-IP experimental work, and co-wrote the manuscript. G.D. G. performed automated and semi-automated protein intensity quantification at centrosome, nucleus/cytoplasm and protrusions. Y.Y.Z. generated the Tet-inducible stable cell lines, prepared ACM and provided bladder cancer cell lines. A.A.O. generated and purified bacterially expressed PLK4 and AURKB and conducted GST pull-down assays. M.Ba. wrote the scripts for automated random selection of ROIs in cells that failed to form protrusions, and cell aspect ratio quantification. J.M.T. contributed to generate and validate phospho-PLK4 (S305) and phospho-PLK4 (T170) antibodies, and reviewed the paper. A.Q. U. prepared ACM and purified exosomes from L cells. L.Z. provided the breast cancer cell lines and prepared ACM. M.Bo. helped with WB validation. L.P. and J.L.W. provided resources, feedback on experimental design, data analysis and interpretation, and assisted with data analysis and contributed to the writing of the paper.

## Additional information

**Competing interests:** The authors declare no competing interests.

