## [Peer Review File · Nature Communications]

Reviewers' comments:

Reviewer #1 (Remarks to the Author):

In this manuscript, Luo and colleagues show that non-directional motion of cells responded to exosome stimuli is independent of centrosomes or microtubules, but mediated by atypical function of the Cep192/Plk4/AuroraB complex at the cell cortex. Using Cep192RNAi or a combination of centrinone and AZD, they demonstrate that these factors specifically regulate exosome-induced cell motility. It has been reported by these authors that WNT-PCP signaling pathway is involved in this kind of cell motility in the same situation. In this study, they claim that DVL2, a part of WNT-PCP pathway, acts as a hub to form the protein complex at the cell cortex. A LUMIER screen suggests their possible interactions and the authors provided some biochemical results of the physical interactions within the protein complex. The Cep192/Plk4/AuroraB complex is recruited to the cortex of the cells with protrusive activity, which is mediated by the function of DVL2. DVL2-Cep192 module seems to be required for the loading of Plk4 and AuroraB to the cell cortex, and these kinases are mutually independent for their recruitment. However, the ways of DVL2 regulating the loading of these kinases to the cell cortex are somewhat different; DVL2 regulates AuroraB export from the nucleus whereas DVL2 suppresses Plk4 degradation through the interaction with BTRC. As a downstream of these events, the authors finally claim that the change in the localization of DAAM1 and DAAM2 at the cell cortex is critical for the cell motility.

Overall, this is a potentially interesting study that illustrates the novel function of centrosome components in centrosome-independent cancer cell motility. However, this study would not provide a mechanics insight into how these components atypically assemble and this module triggers actin remodeling at the cell cortex. In particular, the following concerns preclude publication in Nature Communications at this stage, and should be addressed.

Major points:

- 1) Fig.4: The biological significance of the interactions in the DVL2-Cep192-Plk4-AuroraB complex is not clear. To support the claim, this should be tested using specific DVL2 deletion mutants compromising each interaction. Although the authors attempted to narrow down the specific binding regions of DVL2 against AuroraB, Plk4 and Cep192 by Co-IP experiments, the mapping was not convincing. They only showed that N or C-terminal half of DVL2 did not bind to Plk4 and AuroraB, presumably leading to a failure in rescuing the cell motility. They could use a LUMIER assay to identify more specific and distinct binding regions of DVL2 to each protein, and address the biological relevance of each interaction in the cell motility. Using such specific mutants of DVL2 in the background of endogenous depletion, they could also test the recruitment of Cep192, AuroraB and Plk4 to the cell cortex.
- 2) Fig.4: The authors indicated an interesting localization pattern of the DVL2-Cep192-Plk4-AuroraB complex upon exosome stimuli. However, evidence showing their colocalization at the cell cortex is missing. To confirm this, the authors should perform detailed analysis on their colocalization at the cell cortex in a quantitative manner after the protrusion of cells. If they act together to control the actin reorganization, they should be colocalized there and the DVL2 recruitment is supposed to precede that of Cep192-AuroraB-Plk4.
- 3) Fig.6: The model that DVL2 regulates accumulation of AuroraB and Plk4 at the cell cortex in different ways is interesting, but the data is not convincing. If the author would like to prove this, the specific mutants of DVL2 like mentioned above should be tested in these experiments.
- 4) Fig.7 and 8: Although lack of DAAM1 rescues the phenotype of Centrinone + AZD is interesting, how Plk4 and AuroraB redundantly regulate the localization of DAAM1 is not clear at all. DAAM1 or its upstream could be a substrate of both kinases? DAAM1 seems a very downstream of what the authors

mention in the previous figures and the direct link between DAAM1 and the kinases based on experiments is missing. This reviewer is not sure that this part is critical for the main claim of this manuscript.

Minor points:

- 1) Fig. 3: Does overexpression of Plk4 or AuroraB or the combination affect the cell motility and protrusion of cells?
- 2) Fig. 4: d and e, the bands of these western blots should be quantified to see the precise binding affinity of DVL2 deletion mutants to AuroraB and Plk4.
- 3) Fig.5: this reviewer is wondering how Cep192 is loaded to the cell cortex from the centrosome. Again, it would be better to test the binding-deficient mutant of DVL2 on this displacement. Otherwise, it should be at least discussed.
- 4) Fig.6: in g, the data is not convincing. Why does low expression of DVL2 increase the interaction between Plk4 and BTRC? Would it be possible that overexpression of DVL2 stabilizes Plk4 by any other means?
- 5) Fig.S7e: gray color for ACM is missing in the columns.

Reviewer #2 (Remarks to the Author):

Review of Atypical Function of Centrosomal Module in WNT-PCP Signalling Drives Contextual Cancer Cell Motility by Luo et al.

This elegant study by Pelletier and colleagues describes a surprising function for the centrosomal scaffolding protein CEP192, PLK4 and the mitotic kinase, Aurora-B in exosome-induced protrusive activity and motility. The data in the manuscript is high quality and the authors present a convincing case for a new regulatory function of this protein module. The manuscript is clearly written and straightforward to follow despite the vast quantities of data included. I believe the paper would be of interest to researchers in several fields.

Active conditioned medium (ACM) derived from cancer-associated fibroblasts has been previously shown to contain exosomes that induce WNT-dependent protrusion formation and migration in cancer cells. In particular, upon entry of exosomes into breast cancer cells, CD81 recruits Wnt11; exosomes containing CD81-Wnt11 are then released from the cancer cells and promote protrusive activity, motility and invasion. The planar polarity pathway (PCP) is responsible for transmitting these signals via FZD and DVL. These findings were originally described in a highly cited publication in Cell (2012) by one of the co-authors, and the current manuscript is largely based on this model system.

The key results of this manuscript are listed below:

- 1) ACM-dependent protrusive activity and motility is independent of microtubules and centrosomes, but requires the centrosomal protein CEP192, the centrosomal kinase PLK4, and the chromosome passenger components, Aurora-B and Aurora-C kinases.
- 2) The PCP protein DVL2 recruits CEP192 to the cell cortex, which in turn localises PLK4 and Aurora-B kinases.
- 3) PLK4 and Aurora-B kinases function redundantly in ACM-driven protrusive activity and motility.
- 4) PLK4 and Aurora-B mediate recruitment of DAAM2 formin to ACM-induced protrusions, a process opposed by DAAM1.
- 5) DAAM2 formin is required for ACM-induced motility

Major points:

1. Although there are examples for interplay between Aurora and Polo kinases, it is difficult to imagine that these two kinases would have evolved redundancy under selective pressure in cancers. It is more feasible that Aurora-B and PLK4 act redundantly in a normal (essential) physiological process (perhaps in gastrulation or tissue remodelling during development) and this network is re-utilised in ACM-induced motility of cancer cells. The authors suggest that this pathway might operate in multiciliated cells, which is of course feasible, but other possibilities should be discussed, too.

2. As the authors discuss there are several publications on cell-autonomous roles of Aurora-B and PLK4 in cell migration, and in pathological cell invasion. These functions could also involve the Wnt signalling/Dvl/CEP192 module, and therefore it would be important to test if CEP192/PLK4/AurB can localise to the cortex in other modes of cell motility. Moreover, it has been shown by Godinho and colleagues that PLK4-induced centrosome amplification can drive cortical motility and invasion, and that this is CEP192-dependent (presumably due to centrosomal microtubule nucleation). Whilst not essential, it would be interesting to learn if in PLK4-overexpressing MCF10A cells CEP192/PLK4/AurB is recruited to the cortex.

3. In Fig. 3e, addition of centrinone+AZD causes a gradual decrease in motility speed over 14 hours. By contrast, cells return to normal speed within 1 hour of drug removal. This slow kinetics is very surprising and it is more in line with the kinases controlling gene expression rather than acting directly on actin cytoskeleton. Could the authors comment on this?

4. An important piece of data seem to be missing. Title of Supp Fig 2 legend states that CEP192 depletion does not disrupt WNT11 trafficking, but this data is not shown in the figure. In my view, it is crucial to demonstrate that not only do exosomes get into CEP192-depleted cancer cells, but that they also recruit WNT11, and exosomes with CD81-WNT11 are secreted from these cells. It is also worth noting that depletion of Prickle1 reduces ACM-induced migration speeds more than CEP192 depletion, and it also causes a reduction in general migration speeds (Fig. 2b).

5. Quality of western blot in Fig. 4b (AurB) is poor and the interaction is far from convincing. The legend states n=3; is this really the best blot of the 3? If so, perhaps the authors should include quantitation. I would also expect to see stronger interactions in ACM-stimulated MDA-MB-231 cells. Is there a technical reason for not showing IPs in ACM-treated cancer cells? Given the frequent use of "contextual" throughout the text (referring to ACM-induced processes) I am surprised that these proteins interact at all in HEK293 cells. Do they go to the cortex in HEK293?

Minor points:

1. In Fig. 1h and Supp Fig 1c; y axis label states % area recovered. It is therefore counterintuitive that recovery is greater in ACM- and centrinone-treated cells that are expected to show poorer recovery from text.

2. "on page 11 it is stated that 960 breast cancer patients deposited in ATCG" This must refer to the TCGA database. In Supp Fig 3f, which method (i.e. HiSeq mRNA or microarrays) is the expression data based on? What threshold was used to define upregulation and no upregulation?

3. It would be helpful to change colours in the graph in Supp Fig 7i could be changed, as in general red marks overexpression, whereas here it refers to decreased expression.

Reviewer #3 (Remarks to the Author):

Summary

Previous data from the authors demonstrate that exosome-dependent autocrine WNT11 signaling stimulates cancer cell motility through a pathway involving Frizzled, Dishevelled, Prickle, and Vangl1. In the current paper the authors explore mechanisms underlying exosome-mediated regulation of actin reorganization and cell migration. Their experiments led them to investigate the roles of multiple centrosomal proteins (PLK4, CEP192, and AURKB) during cancer cell motility and to identify the relationship between these proteins and exosome-WNT-PCP signaling. In this manuscript, it is shown that Dishevelled2 recruits centrosomal proteins to membrane protrusions where PLK4/AURKB promote DAAM2-dependent actin re-organization and cell migration. This manuscript additionally shows correlations between the expression of these proteins and breast and bladder cancer progression. Ultimately, the authors claim that this research identifies an interesting and unexpected connection between centrosomal proteins and WNT-PCP signaling to control cell motility and invasion. Overall I found this manuscript to be interesting and very well written. The conclusions are supported by the data and I commend the authors for their extensive use of controls and proper data analysis including statistics. The level of experimental detail provided for repeatability is no better or worse than is typically found in journals.

Specific comments (major)

-The major criticism I have is the use of the phrases "WNT-PCP signaling" and "WNT-PCP pathway" throughout the manuscript. This implies a pathway regulating the morphogenetic process of planar cell polarity. What the authors are studying is not PCP and this manuscript does not add any data to our knowledge of PCP as a morphogenetic process (and that is OK). The authors are indeed careful with their language and do not state that they are studying a true PCP process. This manuscript and previous work (e.g. Luga et al., 2012) identifies a WNT signaling pathway(s) that uses so-called PCP proteins to regulate cell motility. However, the phrase "planar cell polarity" must be used sparingly or else it is misleading for the reader. In other words, just because a PCP protein is required for a cell behavior, this does not mean that the cell behavior is truly PCP. Goodrich and Strutt (2011) have written an excellent review on this topic. I would simply use the phrase WNT signaling pathway throughout the manuscript.

The data present are very compelling and quite beautiful. However, this manuscript provides data regarding how a WNT signaling pathway (and now centrosomal proteins) influences general in vitro cell motility and not in vivo PCP. The authors just need to be careful with their writing. The discussion section may benefit from having a single paragraph discuss how this manuscript's data might inform our knowledge or apply to true PCP as occurs during vertebrate gastrulation, neural tube closure, and perhaps even the fly eye or wing epithelium.

The last paragraph of the discussion section states "planar cell polarity signaling has been implicated in the control of ciliary beating and ciliogenesis". Again, this is misleading. It would be better to state that PCP proteins have been implicated.

While I appreciate your very nice previous publications, the DVL2 data presented in this manuscript only implicate DVL2 and not VANGL proteins or Prickle etc. I don't think you need to do more experiments, just change the language used in the manuscript as mentioned above.

Specific comments (minor)

-If possible I suggest reducing the number of acronyms used in order to make the manuscript easier to read. There are times when a word is used so infrequently that there is no need for an acronym in the main text, though it will need defining in the figure legend. An example of this is nocodazole (page 9). Also on page 9, the authors introduce acronyms for cytochalasin B and D; however, they also re-

introduce these acronyms in the Figure 1 legend.

-Figures 1, 4, 5, and 7 have panels with either images or text that may be too small for reader's eyes, especially if these figures are further reduced for publication as I expect they will be. For example, Figure 1f, 1g, and 1h; Figure 4d, 4e, and 4h; Figure 5c, 5d, 5e, 5g, 5h, 5i, 5k, 5m, and 5o; Figure 7a, 7b, 7c, the right sides of 7d, 7e, 7f, and 7g, and lastly 7j. At the authors/editors discretion, some figure panels may be moved to Supplementary Data so that figure panel sizes can be enlarged.

-For the siRNA experiments, the methods section states that 20 nM siRNA was used. However, this research used both single siRNAs and pools of four siRNA. Was 20 nM used even for the pools, thus 5 nM of each siRNA in the pool? This just needs clarification.

-In supplementary Figure 3c and 3d, it wasn't clear why qPCR was used to validate the AURKC siRNA rather than western blot. Lack of a good antibody?

-In the Figure 4 legend, performed is spelled preformed.

-On the top of page 14, change "which continuous exchange" to that continuously exchanges

-Last paragraph on page 15 you state "DVL2 may disrupts"

-I strongly recommend changing this statement on page 19 "Little is known about the mechanisms whereby PCP regulates cell migration such as in convergent extension". This is a very broad statement and as such ignores the contribution of many papers to our understanding of PCP during gastrulation. I would make your statement more specific as to our lack of understanding how PCP proteins regulate actin remodeling during gastrulation.

-Page 20 paragraph 1. Do you have any thoughts on how WNT signaling via DVL2 may regulate shuttling of AURKB out of the nucleus?

-I found this sentence to be a little strong given the correlative nature of the cancer patient data. "Our study thus reveals an atypical function for a centrosomal module in WNT-PCP signalling and actin nucleation that is critical for cancer cell motility associated with aggressive forms of human cancer.

-In certain instances you use "signaling" and other "signalling".

-Last paragraph on page 21 you state "preforms" instead of performs.

Reviewer #4 (Remarks to the Author):

The manuscript by Luo et al describes a novel signalling pathway driving non-directional cell motility of single cells, which utilises components of Planar Cell Polarity (PCP) pathways and proteins previously thought to be mostly at centrosomes (e.g. PLK4). The authors use primarily fibroblast-conditioned medium (termed active conditioned medium, ACM) as a stimulus and describe that this stimulates dishevelled2-mediated recruitment of the kinases PLK4 and AURKB/C, in a redundant fashion, to form actin-based protrusions in a DAAM2 formin-dependent fashion.

This work is interesting as it identifies and characterises the mechanisms of a novel, centrosome-independent form of cell motility. I want to state my enthusiasm and support for this work at the outset. However, the manuscript is plagued by overstatements, inaccuracies, misleading representations of data, and lack of clarity surrounding several experiments. I think that if these unnecessary overextensions were removed, or clarified, then this work would have only enthusiastic support. I detail these below.

The authors need to be much clearer about which specific conditions are used, and not mischaracterise what was done. For instance, '...exosome-treated cells' are referred to in the manuscript in various places, but figure legends detail that experiments actually use conditioned-

medium treated cells. This is misleading, so please correct.

The authors describe Wnt-PCP signalling, but do not show that Wnt from the conditioned medium confers the presented effects. Wnt was identified as regulating cell behaviour in previous work from Wrana et al. That does not mean that it is Wnt (eg. Wnt11) controlling this novel pathway. The authors need to do such experiments, or soften claims.

Similarly, the authors describe this pathway as 'Wnt-PCP'. There is no demonstration of Wnt involvement. Similarly, these cells do not have PCP, so how can this be PCP signalling? This is akin to renaming anything that involves an FGF receptor as 'FGF-body axis signalling' because FGF has been involved in regulating laterality in other contexts. Just as PLK4 is described here as NOT being centrosomal signalling, surely this should NOT be PCP signalling, without direct demonstration of a function in actual PCP.

To this end, how does the phenotypes and signalling pathways described here fit in the context of previous work from Wrana et al, describing a Wnt11-Fzd6-Dvl pathway controlling protrusions via Vangl/Prickle and ARHGAPs 21/23 and RhoA? Some discussion of this would be very helpful.

Regarding the Supplementary Movies. The authors need to better describe what the movies actually show. The description at the start of page 8 could be misconstrued that they are showing effects 'in a variety of BCC lines'.

MDA-MB-231 are used in most experiments. These are highly heterogenous cells, with some round and some very elongated. Changes to cell shape are shown in single images, such as being round then transitioning to have protrusions. It is thus important to show the penetrance of these changes to shape in such a mixed morphology cell line, with quantitation associated with every figure wherein a different modulation was performed.

Regarding Fig. 1f, it is difficult to understand how the authors conclude that ACM stimulates random walk from the data presented. The authors should more clearly explain this.

Regarding Fig 1g,h – relating to this figure and to the others where a similar incidence occurs, the authors should state that differences might be significant, but that the magnitude is quite small. Otherwise this seems overstated.

As a general comment, many of the images are extremely dim and hard to appreciate. Certainly, much of the images were not visible in printed copies.

Regarding Fig 2a, and Fig S2a. Provide quantitation of these phenotypes. This is particularly important for the 3D culture, as it is unclear how the conclusions of less protrusive activity can be supported from the provided images. I can find in these alternate images alternate regions of interest to show different effects than those claimed. Without quantitation, these are selective and unscientific.

For statistical analyses throughout, there is a lack of consistency. The authors need to state to which conditions statistical analyses are compared. For example, in Fig 3a, presumably the control of statistical analysis is the second box plot, DMSO+ACM. In other figures, the first box plot is what conditions are compared to DMEM alone. Be consistent and provide a clear explanation in the main text as to why statistical analyses are not applied statistically in the main text body. Otherwise it is not clear between figures exactly what is being compared to what.

Regarding Fig S3w, the position of the significance-indicating asterisks is mis-positioned, and hard to see.

Regarding Fig S4, why is there a lack of bidirectionality in the LUMIER approach? For instance, PLK4

fails to fish out ARKB/C, but these fish out PLK4. Similarly, between PLK4 and CEP192? This is not mentioned.

Regarding Fig 4 and S4. There is a wide variation in input expression of a number of these exogenously expressed proteins. Provide quantitation of these experiments performed in triplicate, so to ascertain whether these are really effects on stability, as claimed by authors, or whether these are simply experimental variation or transfection efficiency issues.

The authors describe (Fig 4) that Dvl2 seems to be controlling 'steady-state' levels of PLK4. However, all of these experiments are performed with exogenously expressed protein. Does Dvl2 depletion or overexpression change actual steady-state levels (i.e. endogenous levels) of PLK4 or AURKB/C? The authors present Dvl2 KD and OX, so it is curious why this is not provided.

Regarding the quantitation of cortical levels of protein, from Fig5 onwards. It is inappropriate to use a method of 'cortical' recruitment quantitation wherein the definition of the cortex is changed between conditions. Specifically, to compare the entire cortex in one condition, versus only a protrusion in another condition is unscientific and misleading. I understand that the authors are attempting to compare specialised functional zone (protrusion) versus unspecialised steady state (general cortex). However, these are not equivalent. Either compare sampling of the whole cortex in all conditions, or equally sized subzone vs equally sized subzone.

Regarding Fig 6g,h. The authors are positing that DVL2 inhibits BTRC binding to PLK4, thereby likely leading to PLK4 stabilisation. If this is the case, why does overexpressing lower and medium levels of exogenous Dvl2 initially stabilize the PLK4-BTRC interaction, and lead to increased PLK4 levels? It is only when very high levels DVL2 are expressed is the BTRC-PLK4 levels disrupted. Moreover, the levels of PLK4 appear the same, whether or not its association with BTRC is disrupted. The data do not therefore support such a simple model.

The discussion is overly long and mostly a rehash of the data. It would be very helpful to have a more concise summary of how the presented data fits in with the current literature, and the previous pathways identified by these groups.

We would like to thank the referees for their enthusiasm about our work, judicious comments and the many thoughtful suggestions. Below is our detailed point-by point response (our responses are in bold and the original comments in their entirety are in *italic*). The referee reports have been very helpful and we hope that the reviewers will now find the revised version of our manuscript suitable for publication in *Nature Communications*.

Reviewer #1 (Remarks to the Author):

In this manuscript, Luo and colleagues show that non-directional motion of cells responded to exosome stimuli is independent of centrosomes or microtubules, but mediated by atypical function of the Cep192/Plk4/AuroraB complex at the cell cortex. Using Cep192RNAi or a combination of centrinone and AZD, they demonstrate that these factors specifically regulate exosome-induced cell motility. It has been reported by these authors that WNT-PCP signaling pathway is involved in this kind of cell motility in the same situation. In this study, they claim that DVL2, a part of WNT-PCP pathway, acts as a hub to form the protein complex at the cell cortex. A LUMIER screen suggests their possible interactions and the authors provided some biochemical results of the physical interactions within the protein complex. The Cep192/Plk4/AuroraB complex is recruited to the cortex of the cells with protrusive activity, which is mediated by the function of DVL2. DVL2-Cep192 module seems to be required for the loading of Plk4 and AuroraB to the cell cortex, and these kinases are mutually independent for their recruitment. However, the ways of DVL2 regulating the loading of these kinases to the cell cortex are somewhat different; DVL2 regulates AuroraB export from the nucleus whereas DVL2 suppresses Plk4 degradation through the interaction with BTRC. As a downstream of these events, the authors finally claim that the change in the localization of DAAM1 and DAAM2 at the cell cortex is critical for the cell motility.

*Overall, this is a potentially interesting study that illustrates the novel function of centrosome components in centrosome-independent cancer cell motility. However, this study would not provide a mechanics insight into how these components atypically assemble and this module triggers actin remodeling at the cell cortex. In particular, the following concerns preclude publication in *Nature Communications* at this stage, and should be addressed.*

We thank this reviewer for suggesting to further dissect the DVL2/CEP192/AURKB/PLK4 recruitment hierarchy at the cell cortex in response to exosomes. We have now explored this further as detailed below.

Major points:

1) Fig.4: The biological significance of the interactions in the DVL2-Cep192-Plk4-AuroraB complex is not clear. To support the claim, this should be tested using specific DVL2 deletion mutants compromising each interaction. Although the authors attempted to narrow down the specific binding regions of DVL2 against AuroraB, Plk4 and Cep192 by Co-IP experiments, the mapping was not convincing. They only showed that N or C-terminal half of DVL2 did not bind to Plk4 and AuroraB, presumably leading to a failure in rescuing the cell motility. They could use a LUMIER assay to identify more specific and distinct binding regions of DVL2 to each protein, and address the biological relevance of each interaction in the cell motility. Using such

specific mutants of DVL2 in the background of endogenous depletion, they could also test the recruitment of Cep192, AuroraB and Plk4 to the cell cortex.

The reviewer raises a good point in suggesting that the biological significance of the interactions between components of the DVL2-CEP192-PLK4/AURKB module are not completely clear. In the first submission, we used N- and C-terminal truncations and previously known domains of DVL2¹ to define their requirement in associating with AURKB and PLK4. We had observed that either the N- or C-terminal halves of DVL2 were required for PLK4 and AURKB binding, indicating that large portions of DVL2 are necessary for the association (Supplementary Figs. 5a and 5b). We also further quantified the DVL2 domain mapping with AURKB/PLK4 by LUMIER and validated the co-IP data (Supplementary Figs. 5d and 5e), which shows that either the N- or C- halves are required for DVL2 association with AURKB/PLK4. As included in our first submission, we noticed that some of the deletion mutants tested were poorly expressed/unstable (we have now quantified this and include the data in Figs. R1-1d and R1-1f). For example, the expression level of the Δ DEP mutant is only ~5% of wild type DVL2 (Fig. R1-1f), thus making it difficult for us to ascertain binding efficiency of such mutants and this reason precluded us from further mapping the precise binding domains between DVL2 and other module components. Having said this, we agree that it is important to better understand the binding relationship between DVL2 and CEP192-PLK4/AURKB and so we focused our efforts on CEP192 which gives conclusive results in our hands. We now mention issues encountered expressing these domains in the methods section of the revised manuscript on page 46.

We previously observed that DVL2 depletion led to a loss of CEP192 at the cortex (Figs. 5a and 5b), and either DVL2 or CEP192 depletion led to loss of AURKB and PLK4 in protrusions (Figs. 7a-7d). We have now mapped the DVL2-CEP192 association domain. Our results show a similar binding pattern as we observed for DVL2-AURKB and DVL2-PLK4, with both the N- and C-terminal halves of DVL2 being required for CEP192 association (Supplementary Fig 5c). More importantly, we now show that depletion of CEP192 significantly reduces the interaction between DVL2 and PLK4/AURKB (Supplementary Fig. 8a), indicating that DVL2 association with the kinases depends on CEP192. We therefore hypothesized that DVL2 interacts with AURKB/PLK4 via CEP192. To test this hypothesis, and as suggested by this reviewer, we set out to further explore the binding relationship between CEP192 and PLK4/AURKB and its role in exosome-stimulated cell motility. For this we took advantage of findings from previous work indicating that only the longest isoform of CEP192 (1-2537, isoform-1) interacts with PLK4 at AA190-240². We now show that both the long (1-2537) and short isoform-2 (1-1941) of CEP192 interact with DVL2 and AURKB, but that the long isoform can only associate with PLK4, thus confirming that CEP192 indeed recruits PLK4 to DVL2 (Supplementary Fig. 8b). Furthermore, using a previously published siRNA specific to the long CEP192 isoform-1^{13,4} (Supplementary Fig. 8c), we now show that ACM still induces increased BCC motility (Supplementary Fig. 8d), but now AZD treatment alone inhibits BCC migration. Consistent with our model, knockdown of the long CEP192 isoform-1 resulted in failure to recruit PLK4 to protrusions, with no effect on the recruitment of other CEP192 isoforms and AURKB (Supplementary Fig. 8e-8j).

These results suggest that when the long isoform-1 of CEP192 is absent, the short isoforms can only recruit AURKB, thus making the pathway to become sensitive to just AURKB inhibitors (Supplementary Fig. 8d). In conclusion, these findings further support our conclusions that DVL2 is required for the recruitment of CEP192 to protrusions, and now show that while CEP192 isoform-1 recruits both PLK4 and AURKB, the non-PLK4 binding short CEP192 isoforms still recruit AURKB.

We realize that these new data do not fully address the reviewer's concerns due to the fact that DVL2 does not behave well in our interaction mapping assays. We do hope, however, that the additional data we now provide indicating that both halves of DVL2 are required for CEP192 binding, that CEP192 acts as a bridging component between DVL2 and the kinases, and the separation of function of CEP192 isoforms in regards to their ability to bind either/or PLK4/AURKB in ACM-induced cell motility assays this reviewer's concerns.

2) Fig.4: The authors indicated an interesting localization pattern of the DVL2-Cep192-Plk4-AuroraB complex upon exosome stimuli. However, evidence showing their colocalization at the cell cortex is missing. To confirm this, the authors should perform detailed analysis on their colocalization at the cell cortex in a quantitative manner after the protrusion of cells. If they act together to control the actin reorganization, they should be colocalized there and the DVL2 recruitment is supposed to precede that of Cep192-AuroraB-Plk4.

This is a good suggestion. Due to currently available antibodies and species limitations in co-labeling conditions, it was not possible for us to undertake detailed colocalization analyses of the endogenous proteins. To circumvent this issue, we generated an inducible, FLAG-tagged DVL2, MDA-MB-231 cell line that could be used in pair-wise co-localization analyses with other antibodies. We first confirmed the expression of DVL2 upon Tet induction (Supplementary Fig. 7a). To measure the colocalization between DVL2 (FLAG) and the proteins of interest (POI) (e.g. CEP192, pPLK4S305 and pAURKBT232) at protrusion, we used a colocalization-based algorithm detailed in Supplementary Fig. 7b. Briefly, after defining the cortical region using the actin signal, protrusions were selected as described in the methods section. At protrusions, masks for both DVL2 (FLAG) and POI were generated, and then combined to create a combined mask of two channels. The correlation coefficient between DVL2 (FLAG) and POI was measured under such masks. Finally, the correlation coefficient was normalized to a "positive control", where DVL2 inducible MDA-MB-231 cells were treated with Tetracycline in the presence of ACM and subsequently stained with two different secondary antibodies against the same FLAG primary antibody. Our results indicate that colocalization between DVL2 (FLAG) and CEP192-pPLK4S305-pAURKBT232 was significantly increased at protrusions compared to random signals. These results are now discussed on page 15-16. The order of recruitment was detailed in the original version of the manuscript with DVL2 depletion, and CEP192 being recruited to the cortex prior to PLK4 and AURKB (Fig. 5 and Fig. 7). This being further substantiated now with the analysis of CEP192 isoforms in PLK4 and AURKB recruitment as detailed above (Supplementary Fig. 8).

3)Fig.6: The model that DVL2 regulates accumulation of AuroraB and Plk4 at the cell cortex in different ways is interesting, but the data is not convincing. If the author would like to prove this, the specific mutants of DVL2 like mentioned above should be tested in these experiments.

This is also a good suggestion. As mentioned above, we have not been able to more reliably map the DVL2 interacting domains. However, we have now investigated the interaction between CEP192 isoforms and PLK4/AURKB, since our results indicate that ACM-induced DVL2-dependent recruitment of PLK4 or AURKB to protrusions occurs via different CEP192 isoforms. We now show that upon knockdown of the CEP192 isoform (isoform-1) that specifically binds PLK4, cells still respond to ACM stimulation in an AURKB dependent manner (Supplementary Fig. 8d). Also, PLK4 recruitment to protrusions was inhibited when CEP192 isoform-1 was downregulated, although AURKB accumulation at protrusions remained (Supplementary Figs. 8e-8j). This is now discussed in more detail on page 16-17 of the revised manuscript.

4)Fig.7 and 8: Although lack of DAAM1 rescues the phenotype of Centrinone + AZD is interesting, how Plk4 and AuroraB redundantly regulate the localization of DAAM1 is not clear at all. DAAM1 or its upstream could be a substrate of both kinases? DAAM1 seems a very downstream of what the authors mention in the previous figures and the direct link between DAAM1 and the kinases based on experiments is missing. This reviewer is not sure that this part is critical for the main claim of this manuscript.

We agree with the reviewer that at this stage we don't know the exact mechanism underpinning how PLK4 and AURKB regulate DAAM1 activity at cortex. One possibility may be that it is directly regulated by the kinases. We are currently investigating this but don't have an answer yet. In this manuscript we simply claim that DAAM1 downregulation is dependent on PLK4/AURKB activity, and knockdown of DAAM1 can bypass the requirement of the kinases suggesting DAAM1 acts downstream of the kinases. Since we have shown that exosome-induced cell motility is ARP2/3 and SPIRE1 independent, but formin-dependent we felt that it is important to establish which formins are at play in this process and for this reason we would like to keep these data in the manuscript. We have clarified our thoughts on this issue in the manuscript and now discuss it further on page 22-23.

Minor points:

1)Fig. 3: Does overexpression of Plk4 or AuroraB or the combination affect the cell motility and protrusion of cells?

This is an excellent suggestion, since we have shown in breast and bladder cancer patient, that upregulation of PLK4 or AURKB correlates with reduced survival rate and high-grade disease respectively (Supplementary Figs. 3f and 3g). To test this, we generated FLAG-tagged PLK4 and AURKB inducible MDA-MB-231 cell lines and confirmed their expression (Fig. R1-1a). ACM stimulation increased cell motility in both cell lines to a similar extent than WT MDA-MB-231 cells (Fig. R1-1b). However, upon induction of PLK4 or AURKB expression, we observed increased cell motility in the absence of ACM stimulation (Fig. R1-

1b). The increased cell motility was inhibited by the corresponding kinase inhibitor. Overall, our data suggests PLK4 or AURKB expression is sufficient to induce cell migration. We include this result here for the reviewer's perusal but since it may not be essential to the paper we mention it as "data not shown" on page 22 of the revised manuscript.

2) Fig. 4: d and e, the bands of these western blots should be quantified to see the precise binding affinity of DVL2 deletion mutants to AuroraB and Plk4.

We have now quantified the domain mapping data in Supplementary Figs. 5a and 5b from three independent experiments (Figs. R1-1c-f). We first measured the amount of AURKB/PLK4 (HA) we were pulling down, and normalized it to its total expression level. Then we calculated the binding efficiency by normalizing the resulting data to the amount of DVL2-mutant (FLAG signal) being pulled down. The data shows that both N- or C-terminal halves of DVL2 associate far less with AURKB and PLK4 than the full length protein, supporting the notion that these two regions cooperate to bind CEP192 interaction and thus AURKB/PLK4 association (see above, Reviewer 1 Major point 1). We also preformed LUMIER assays, a quantitative method, which validated our co-IP data and also shows that both N- and C- halves are required (Supplementary Figs. 5d and 5e). As mentioned above, we are concerned about our inability to efficiently express some of the DVL2 mutants and make reliable conclusions with them (Figs. R1-1d and R1-1f). Together with the CEP192 data described above (Reviewer 1 Major point 1), we now argue that DVL2 recruits AURKB/PLK4 to protrusions via CEP192.

3) Fig.5: this reviewer is wondering how Cep192 is loaded to the cell cortex from the centrosome. Again, it would be better to test the binding-deficient mutant of DVL2 on this displacement. Otherwise, it should be at least discussed.

Our data indicates that centrosomes are not required for exosome-induced cell motility and we therefore believe that the cytoplasmic pool of CEP192 can relocate to the cortex independently of centrosomes. In the presence of centrosomes, CEP192 levels at the centrosomes are decreased upon exosome stimulation. Given that centrosome-localized CEP192 is known to be in dynamic equilibrium with its cytoplasmic pool¹⁵, we think the most likely possibility is that the cytoplasmic pool of CEP192 relocates to protrusions. As mentioned above, we now show that CEP192 association with DVL2 requires both the N- and C- terminal halves of DVL2 (Supplementary Fig. 5c). This is now further discussed on page 13 of the revised manuscript.

4) Fig.6: in g, the data is not convincing. Why does low expression of DVL2 increase the interaction between Plk4 and BTRC? Would it be possible that overexpression of DVL2 stabilizes Plk4 by any other means?

We thank the reviewer for pointing this out. In the original version of the manuscript we had not quantified the fluctuations in BTRC/PLK4 binding relative to total PLK4 expression levels. In the revised version of the manuscript, we have now quantified data from four independent experiments, and find that upon low-level expression of DVL2 the increase in PLK4/BTRC association, although noticeable higher, is not statistically significant (Fig. 8h).

We believe this small increase may be due to a concomitant increase of overall PLK4 amount under low DVL2 expression. We can't however, at this stage, exclude the possibility that other mechanisms may be involved in stabilizing PLK4 levels under overexpression of DVL2. For instance, protein phosphatase 2A (PP2A) has been shown to stabilize PLK4⁶, and directly interacts with DVL2⁷. Therefore, it is possible that low expression of DVL2 recruits PP2A to the DVL2/PLK4 module which in turn can contribute to stabilize PLK4. We now discussed this possibility further on page 18 of the revised manuscript.

5) Fig.S7e: gray color for ACM is missing in the columns.

This error has been corrected.

Reviewer #2 (Remarks to the Author):

Review of Atypical Function of Centrosomal Module in WNT-PCP Signalling Drives Contextual Cancer Cell Motility by Luo et al.

This elegant study by Pelletier and colleagues describes a surprising function for the centrosomal scaffolding protein CEP192, PLK4 and the mitotic kinase, Aurora-B in exosome-induced protrusive activity and motility. The data in the manuscript is high quality and the authors present a convincing case for a new regulatory function of this protein module. The manuscript is clearly written and straightforward to follow despite the vast quantities of data included. I believe the paper would be of interest to researchers in several fields.

We thank this reviewer for the kind comments on the quality of our data and the significance of our discoveries.

Active conditioned medium (ACM) derived from cancer-associated fibroblasts has been previously shown to contain exosomes that induce WNT-dependent protrusion formation and migration in cancer cells. In particular, upon entry of exosomes into breast cancer cells, CD81 recruits Wnt11; exosomes containing CD81-Wnt11 are then released from the cancer cells and promote protrusive activity, motility and invasion. The planar polarity pathway (PCP) is responsible for transmitting these signals via FZD and DVL. These findings were originally described in a highly cited publication in Cell (2012) by one of the co-authors, and the current manuscript is largely based on this model system.

The key results of this manuscript are listed below:

- 1) *ACM-dependent protrusive activity and motility is independent of microtubules and centrosomes, but requires the centrosomal protein CEP192, the centrosomal kinase PLK4, and the chromosome passenger components, Aurora-B and Aurora-C kinases.*
- 2) *The PCP protein DVL2 recruits CEP192 to the cell cortex, which in turn localises PLK4 and Aurora-B kinases.*
- 3) *PLK4 and Aurora-B kinases function redundantly in ACM-driven protrusive activity and motility.*
- 4) *PLK4 and Aurora-B mediate recruitment of DAAM2 formin to ACM-induced protrusions, a process opposed by DAAM1.*
- 5) *DAAM2 formin is required for ACM-induced motility*

Major points:

1. *Although there are examples for interplay between Aurora and Polo kinases, it is difficult to imagine that these two kinases would have evolved redundancy under selective pressure in cancers. It is more feasible that Aurora-B and PLK4 act redundantly in a normal (essential) physiological process (perhaps in gastrulation or tissue remodelling during development) and this network is re-utilised in ACM-induced motility of cancer cells. The authors suggest that this pathway might operate in multiciliated cells, which is of course feasible, but other possibilities should be discussed, too.*

The reviewer raises a very interesting point that PLK4 and AURKB may also act redundantly in a normal physiological process. For instance, Wnt-PCP signaling is essential during embryonic development, such as during gastrulation and neurulation. It is possible that DVL2/CEP192/PLK4/AURKB module is also used in these processes, and we now further discuss this on page 23-24, *“Furthermore, disruption of PLK4 function leads to photoreceptor orientation defects in frog somite and eye implying a role in PCP pathway⁸. Although AURKB-/- embryos are normally implanted, consistent with our findings, AURKC replaces AURKB function during embryogenesis⁹, and therefore the importance of AURKB in embryonic development may still be underappreciated. Indeed, in zebrafish that lack an AURKC ortholog, AURKB localizes within motor neurons in the developing spinal cord and regulates axonal outgrowth and guidance¹⁰, which is known to be mediated by the WNT-PCP pathway. Therefore, it will be intriguing to further investigate the role of AURKB/C and PLK4 in human embryo/tissue development, especially if there are redundant functions that remain to uncovered.”*

2. As the authors discuss there are several publications on cell-autonomous roles of Aurora-B and PLK4 in cell migration, and in pathological cell invasion. These functions could also involve the Wnt signalling/Dvl/CEP192 module, and therefore it would be important to test if CEP192/PLK4/AurB can localise to the cortex in other modes of cell motility. Moreover, it has been shown by Godinho and colleagues that PLK4-induced centrosome amplification can drive cortical motility and invasion, and that this is CEP192-dependent (presumably due to centrosomal microtubule nucleation). Whilst not essential, it would be interesting to learn if in PLK4-overexpressing MCF10A cells CEP192/PLK4/AurB is recruited to the cortex.

We thank the reviewer for this suggestion. We have now tested the localization of CEP192/PLK4/AURKB during directional migration (wound healing assays) in MCF10A. As suggested by this reviewer, we first transiently transfected MCF10A with GFP-PLK4 to mimic the PLK4 overexpression condition reported by Godinho and colleagues. We then performed wound healing assays to stimulate directional migration, and fixed the cells 3h after. We did not observe any accumulation of activated PLK4/AURKB or CEP192 at leading edge (Fig. R2-1c). This is not consistent with previous findings indicating that PLK4 localizes at leading edge in MEFs and Hela cells¹¹. However, we believe that PLK4 may function differently in directional versus non-directional migration in different cell types. It has been reported that PLK4 regulates directional migration via the ARP2/3-dependent pathway¹², which is dispensable in ACM-induced, non-directional migration that rely on formins (Fig. 9a). Also, we suggest that ACM stimulation of BCC non-directional migration competes with directional migration (Fig. 1g) and we therefore believe that the CEP192/PLK4/AURKB module is specifically stimulated through WNT signalling during non-directional migration. Overall, these data suggest PLK4 is functionally required in both directional and non-directional migration, but its function can differ based on extracellular signals. Excitingly, they further support the notion that extracellular cues can drive contextual control of varied motile systems. This is now further discussed on page 21-22 of the manuscript.

3. In Fig. 3e, addition of centrione+AZD causes a gradual decrease in motility speed over 14 hours. By contrast, cells return to normal speed within 1 hour of drug removal. This slow kinetics is very surprising and it is more in line with the kinases controlling gene expression rather than acting directly on actin cytoskeleton. Could the authors comment on this?

This reviewer makes an interesting point. Although our data suggests that PLK4 and AURKB activity are required for DAAM1 removal from the cell cortex, we don't know if this is a direct or indirect process. It is thus possible that DAAM1 removal occurs via transcriptional regulation. To test this, we inhibited transcription in BCC using actinomycin D, and investigated its effect on cell motility upon ACM stimulation. Our results illustrated that blocking transcription inhibited ACM-induced BCC migration (Fig. R2-1b). Moreover, Actinomycin D takes about 12 hours to fully inhibit BCC migration, which is similar to what we observed using centrione and AZD1152 dual inhibition (Fig. 3e and Fig R2-1c). However, nonlinear regression analysis (Plateau followed by one phase decay) of these data indicates a much faster decrease (half life = ~1.7 hour) in centrione and AZD1152 double inhibition compared to Actinomycin D (half life = 5.5 hour), which suggests that PLK4/AURKB have a direct function in cell motility independent of transcriptional regulation. This will require further investigation, but we now mention these data as “data not shown” in the revised manuscript on page 11-12 due to space constraints.

4. An important piece of data seem to be missing. Title of Supp Fig 2 legend states that CEP192 depletion does not disrupt WNT11 trafficking, but this data is not shown in the figure. In my view, it is crucial to demonstrate that not only do exosomes get into CEP192-depleted cancer cells, but that they also recruit WNT11, and exosomes with CD81-WNT11 are secreted from these cells. It is also worth noting that depletion of Prickle1 reduces ACM-induced migration speeds more than CEP192 depletion, and it also causes a reduction in general migration speeds (Fig. 2b).

We apologize for this. We took out these data in the original manuscript at the last minute due to space concerns and forgot to remove mention of it from the figure legend. We have now included these data in the revised version of the manuscript showing that CEP192 does not affect exosome uptake and processing by BCC (Supplementary Figs. 2f and 2g).

5. Quality of western blot in Fig. 4b (AurB) is poor and the interaction is far from convincing. The legend states n=3; is this really the best blot of the 3? If so, perhaps the authors should include quantitation. I would also expect to see stronger interactions in ACM-stimulated MDA-MB-231 cells. Is there a technical reason for not showing IPs in ACM-treated cancer cells? Given the frequent use of “contextual” throughout the text (referring to ACM-induced processes) I am surprised that these proteins interact at all in HEK293 cells. Do they go to the cortex in HEK293?

We routinely use HEK293T as a model system to study protein-protein interactions (PPIs) since it is a very well established system for LUMIER and BioID^{13, 14, 15, 16}. Moreover, the HEK293T overexpression system has proven to be a reliable way to validate PPIs found by LUMIER¹⁵. But to address the reviewer's point directly, we now show purified GST-AURKB, but not GST alone, can pull down endogenous DVL2 from MDA-MB-231 cell extracts (Fig.

4c). In addition, we agree with this reviewer that to use the expression “contextual” we need to demonstrate the interaction in MDA-MB-231 in the context of ACM stimulation. Therefore, we now provide data indicating that FLAG-tagged DVL2 colocalizes with endogenous CEP192/PLK4/AURKB at protrusion after ACM stimulation (Supplementary Figs. 7c-7g). These data indicate that at protrusions after ACM stimulation, CEP192/PLK4/AURKB are recruited in a DVL2-dependent manner (Fig. 5 and Fig. 7) to DVL2 positive foci (Supplementary Figs. 7c-7g).

Minor points:

1. *In Fig. 1h and Supp Fig 1c; y axis label states % area recovered. It is therefore counterintuitive that recovery is greater in ACM- and centrinone-treated cells that are expected to show poorer recovery from text.*

We apologize for the mislabelling in Fig. 1g and Supplementary Fig. 1c. The control (blue) and ACM/exosome (red) label was reversed and we have now corrected this.

2. *“on page 11 it is stated that 960 breast cancer patients deposited in ATCG” This must refer to the TCGA database. In Supp Fig 3f, which method (i.e. HiSeq mRNA or microarrays) is the expression data based on? What threshold was used to define upregulation and no upregulation?*

We refer to the TCGA database on page 12 and have corrected the typo. The “gene increased expression” refer to DNA copy-number amplification and gain, and mRNA expression (RNA seq V2 RSEM, z-score ≥ 1.7) upregulation. This is now indicated in the figure legend.

3. *It would be helpful to change colours in the graph in Supp Fig 7i could be changed, as in general red marks overexpression, whereas here it refers to decreased expression.*

Unfortunately, the graph was directly generated by TCGA, which defines the altered genes (either upregulation or downregulation) as red. We cannot easily change the color from the plotting data.

Reviewer #3 (Remarks to the Author):

Summary

Previous data from the authors demonstrate that exosome-dependent autocrine WNT11 signaling stimulates cancer cell motility through a pathway involving Frizzled, Dishevelled, Prickle, and Vangl1. In the current paper the authors explore mechanisms underlying exosome-mediated regulation of actin reorganization and cell migration. Their experiments led them to investigate the roles of multiple centrosomal proteins (PLK4, CEP192, and AURKB) during cancer cell motility and to identify the relationship between these proteins and exosome-WNT-PCP signaling. In this manuscript, it is shown that Dishevelled2 recruits centrosomal proteins to membrane protrusions where PLK4/AURKB promote DAAM2-dependent actin re-organization and cell migration. This manuscript additionally shows correlations between the expression of these proteins and breast and bladder cancer progression. Ultimately, the authors claim that this research identifies an interesting and unexpected connection between centrosomal proteins and WNT-PCP signaling to control cell motility and invasion. Overall I found this manuscript to be interesting and very well written. The conclusions are supported by the data and I commend the authors for their extensive use of controls and proper data analysis including statistics. The level of experimental detail provided for repeatability is no better or worse than is typically found in journals.

We thank this reviewer for the kind comments on the quality and significance of our manuscript.

Specific comments (major)

1-The major criticism I have is the use of the phrases "WNT-PCP signaling" and "WNT-PCP pathway" throughout the manuscript. This implies a pathway regulating the morphogenetic process of planar cell polarity. What the authors are studying is not PCP and this manuscript does not add any data to our knowledge of PCP as a morphogenetic process (and that is OK). The authors are indeed careful with their language and do not state that they are studying a true PCP process. This manuscript and previous work (e.g. Luga et al., 2012) identifies a WNT signaling pathway(s) that uses so-called PCP proteins to regulate cell motility. However, the phrase "planar cell polarity" must be used sparingly or else it is misleading for the reader. In other words, just because a PCP protein is required for a cell behavior, this does not mean that the cell behavior is truly PCP. Goodrich and Strutt (2011) have written an excellent review on this topic. I would simply use the phrase WNT signaling pathway throughout the manuscript.

We fully agree this will be a more precise way of describing our work. We now use “WNT signalling” and “PCP protein” as suggested by this reviewer as well as by Goodrich and Strutt (2011) review¹⁷.

2-The data present are very compelling and quite beautiful. However, this manuscript provides data regarding how a WNT signaling pathway (and now centrosomal proteins) influences general in vitro cell motility and not in vivo PCP. The authors just need to be careful with their writing. The discussion section may benefit from having a single paragraph discuss how this

manuscript's data might inform our knowledge or apply to true PCP as occurs during vertebrate gastrulation, neural tube closure, and perhaps even the fly eye or wing epithelium.

We appreciate the suggestion and now discuss the possible roles of this module in PCP on page 23-24 in the discussion section.

3-The last paragraph of the discussion section states "planar cell polarity signaling has been implicated in the control of ciliary beating and ciliogenesis". Again, this is misleading. It would be better to state that PCP proteins have been implicated.

We have corrected this.

4-While I appreciate your very nice previous publications, the DVL2 data presented in this manuscript only implicate DVL2 and not VANGL proteins or Prickle etc. I don't think you need to do more experiments, just change the language used in the manuscript as mentioned above.

Agreed, we have now corrected this.

Specific comments (minor)

1-If possible I suggest reducing the number of acronyms used in order to make the manuscript easier to read. There are times when a word is used so infrequently that there is no need for an acronym in the main text, though it will need defining in the figure legend. An example of this is nocodazole (page 9). Also on page 9, the authors introduce acronyms for cytochalasin B and D; however, they also re-introduce these acronyms in the Figure 1 legend.

We have reduced the number of acronyms.

2-Figures 1, 4, 5, and 7 have panels with either images or text that may be too small for reader's eyes, especially if these figures are further reduced for publication as I expect they will be. For example, Figure 1f, 1g, and 1h; Figure 4d, 4e, and 4h; Figure 5c, 5d, 5e, 5g, 5h, 5i, 5k, 5m, and 5o; Figure 7a, 7b, 7c, the right sides of 7d, 7e, 7f, and 7g, and lastly 7j. At the authors/editors discretion, some figure panels may be moved to Supplementary Data so that figure panel sizes can be enlarged.

We appreciate the suggestion. We have rearranged the figure in the revised version (we now have 9 main figures and 10 supplementary figures) to help with readability.

3-For the siRNA experiments, the methods section states that 20 nM siRNA was used. However, this research used both single siRNAs and pools of four siRNA. Was 20 nM used even for the pools, thus 5 nM of each siRNA in the pool? This just needs clarification.

We used 20nM for all the transfection, and therefore in pool of four siRNA, each siRNA is 5nM. This is now clearly indicated in the methods section.

4-In supplementary Figure 3c and 3d, it wasn't clear why qPCR was used to validate the AURKC siRNA rather than western blot. Lack of a good antibody?

The reviewer is correct. We could not find a reliable AURKC antibody on the market and had to rely on qPCR in that particular case.

5-In the Figure 4 legend, performed is spelled preformed.

This has been corrected.

6-On the top of page 14, change "which continuous exchange" to that continuously exchanges

We have made the requested change.

7-Last paragraph on page 15 you state "DVL2 may disrupts"

This has been corrected.

8-I strongly recommend changing this statement on page 19 "Little is known about the mechanisms whereby PCP regulates cell migration such as in convergent extension". This is a very broad statement and as such ignores the contribution of many papers to our understanding of PCP during gastrulation. I would make your statement more specific as to our lack of understanding how PCP proteins regulate actin remodeling during gastrulation.

This point is well taken and have made the suggested correction.

9-Page 20 paragraph 1. Do you have any thoughts on how WNT signaling via DVL2 may regulate shuttling of AURKB out of the nucleus?

We indeed have some thoughts on this we now include: "Although the precise mechanism of how ACM induces DVL2-dependent AURKB nuclear export remains unclear, DVL2 is known to shuttle between nucleus and cytoplasm¹⁸. DVL2 contains a conserved nuclear export sequence and a nuclear localization sequence, and its nuclear localization is required in the canonical WNT signalling pathway¹⁹. Interestingly, a recent study demonstrated that mutating DVL2 nuclear export sequence results in nuclear YAP accumulation²⁰. It will be intriguing to further investigate if DVL2 also directly regulates AURKB nuclear export though a similar mechanism." This is now discussed on page 17 of the revised manuscript.

10-I found this sentence to be a little strong given the correlative nature of the cancer patient data. "Our study thus reveals an atypical function for a centrosomal module in WNT-PCP signalling and actin nucleation that is critical for cancer cell motility associated with aggressive forms of human cancer.

Agreed. We have now softened our statement on page 23 to "Our study thus reveals an atypical function for a centrosomal module in WNT signalling and actin nucleation that is critical for

cancer cell motility. More importantly, deregulation of genes in this novel module including PLK4, AURKB, DAAM1 and DAAM2 are correlated with aggressive forms of human cancer (Supplementary Fig. 3f and 3g, Supplementary Fig. 10i and 10j)."

11-In certain instances you use "signaling" and other "signalling".

We now use "signalling" throughout.

12-Last paragraph on page 21 you state "preforms" instead of performs.

This has been corrected.

Reviewer #4 (Remarks to the Author):

The manuscript by Luo et al describes a novel signalling pathway driving non-directional cell motility of single cells, which utilises components of Planar Cell Polarity (PCP) pathways and proteins previously thought to be mostly at centrosomes (e.g. PLK4). The authors use primarily fibroblast-conditioned medium (termed active conditioned medium, ACM) as a stimulus and describe that this stimulates dishevelled2-mediated recruitment of the kinases PLK4 and AURKB/C, in a redundant fashion, to form actin-based protrusions in a DAAM2 formin-dependent fashion.

This work is interesting as it identifies and characterises the mechanisms of a novel, centrosome-independent form of cell motility. I want to state my enthusiasm and support for this work at the outset. However, the manuscript is plagued by overstatements, inaccuracies, misleading representations of data, and lack of clarity surrounding several experiments. I think that if these unnecessary overextensions were removed, or clarified, then this work would have only enthusiastic support. I detail these below.

We thank this reviewer for his/her comments on the significance of this work and many good suggestions to improve the quality of our manuscript.

1-The authors need to be much clearer about which specific conditions are used, and not mischaracterise what was done. For instance, ‘...exosome-treated cells’ are referred to in the manuscript in various places, but figure legends detail that experiments actually use conditioned-medium treated cells. This is misleading, so please correct.

This point is well taken, and we did not intend to be misleading. We have modified the manuscript and use “ACM-induced”, except in the experiments where purified exosomes were used.

2-The authors describe Wnt-PCP signalling, but do not show that Wnt from the conditioned medium confers the presented effects. Wnt was identified as regulating cell behaviour in previous work from Wrana et al. That does not mean that it is Wnt (eg. Wnt11) controlling this novel pathway. The authors need to do such experiments, or soften claims.

We’ve now changed our description to “WNT signalling” as suggested by reviewer 3 (see above). Moreover, in the Luga et al paper²¹, we demonstrated that autocrine WNT11 from BCC is required for BCC protrusion formation and high motility via Wnt-PCP components, as knockdown of WNT11 in BCC inhibits ACM-induced BCC migration²¹. In addition, we now include data indicating that the amount of BCC endogenous WNT11 that is loaded into CD81 positive exosomes from L cells remains unaffected in CEP192 RNAi-treated cells (Supplementary Figs. 2f and 2g). This data argues that CEP192’s requirement in ACM-stimulated cell motility is not due to effects on WNT11 loading or secretion. Our preferred model that is consistent with our data, is that CEP192 acts downstream of WNT signalling, and is recruited to protrusions by DVL2 (Figs. 5a and 5b), where it then acts to recruit PLK4 and AURKB via its different isoforms to coordinate DAAM activity (Supplementary Fig. 8).

3-Similarly, the authors describe this pathway as 'Wnt-PCP'. There is no demonstration of Wnt involvement. Similarly, these cells do not have PCP, so how can this be PCP signalling? This is akin to renaming anything that involves an FGF receptor as 'FGF-body axis signalling' because FGF has been involved in regulating laterality in other contexts. Just as PLK4 is described here as NOT being centrosomal signalling, surely this should NOT be PCP signalling, without direct demonstration of a function in actual PCP.

We've now changed our wording throughout the manuscript to better describe our data and findings. Please see detailed comments above in response to reviewer 3 (major point 1) who had similar concerns.

4-To this end, how does the phenotypes and signalling pathways described here fit in the context of previous work from Wrana et al, describing a Wnt11-Fzd6-Dvl pathway controlling protrusions via Vangl/Prickle and ARHGAPs 21/23 and RhoA? Some discussion of this would be very helpful.

We appreciate the suggestion. The Vangl/PRICKLE/ARHGAP21/23 pathway we identified earlier localizes to the non-protrusive lateral membrane, and the atypical centrosomal module we characterize here directly locates to the protrusion to promote cell migration. How these two pathways coordinate cell shape dynamics is an interesting question that needs to be further investigated in the future. We now mention this on page 21.

5-Regarding the Supplementary Movies. The authors need to better describe what the movies actually show. The description at the start of page 8 could be misconstrued that they are showing effects 'in a variety of BCC lines'.

This is a good point, and we've now clarified this in the revised manuscript (page 8).

6-MDA-MB-231 are used in most experiments. These are highly heterogenous cells, with some round and some very elongated. Changes to cell shape are shown in single images, such as being round then transitioning to have protrusions. It is thus important to show the penetrance of these changes to shape in such a mixed morphology cell line, with quantitation associated with every figure wherein a different modulation was performed.

The reviewer raises a very good point, as MDA-MB-231 display highly dynamic cell shape changes that we previously documented over time in individual cells^{21, 22}. Our analysis of a variety of metastatic cancer lines shows similar dynamics that is likely a feature of their metastatic phenotype. Indeed, MDA-MB231 cells gain high metastatic potential *in vivo* when co-injected with fibroblasts in mouse orthotopic models^{21, 22}. Moreover, *in vitro* the stimulated cells show high non-directional motility and continuously form protrusions. As noted above we previously quantified cell shape dynamics in this model and its relationship to motility, so to address this concern, we similarly quantified cell shape in 2D before and after stimulation with ACM. We now provide all the datapoints for the quantification of 2D data throughout the manuscript. Supplementary Figs. 2a and 2b show ACM stimulation

significantly increases the aspect ratio of cell geometry, which reflects protrusive activity. Moreover, knockdown of CEP192 inhibited ACM induced morphological changes, and cells remain round with few or no protrusions formed (Supplementary Figs. 2a and 2b). These data are consistent with our previous findings and indicate that the significant change in cell morphology is associated with ACM stimulation. Moreover knockdown of key component in this atypical centrosomal module, such as CEP192 inhibits protrusion formation, and suppresses cell shape dynamics similar to control non-stimulated cells.

7-Regarding Fig. 1f, it is difficult to understand how the authors conclude that ACM stimulates random walk from the data presented. The authors should more clearly explain this.

We now clarify this in the revised manuscript and the figure legend. We measured the mean square displacement (MSD) of migration traces to measure the spatial extent of random motion. A linear relationship between MSD and time ($R^2=1$) indicates a total random movement, while, for example, migration of cell during wound healing, which is directed would display lower R^2 . Our data demonstrate that MDA-MB-231 cells display random movement in either the absence or presence of exosomes. ACM stimulation simply increases overall speed; but the directionality of the motion is unaffected. This is similar to our previous studies, where we showed ACM affected speed, not direction²².

8-Regarding Fig 1g,h – relating to this figure and to the others where a similar incidence occurs, the authors should state that differences might be significant, but that the magnitude is quite small. Otherwise this seems overstated.

Thank you for pointing this out. We've now included % reduction in the text, which is about 13% decrease at 18h.

9-As a general comment, many of the images are extremely dim and hard to appreciate. Certainly, much of the images were not visible in printed copies.

This is a very good point. We've increased the brightness of the figures. We've also uploaded the high quality versions of the figures which can allow readers to view all the detail of the figures on screen.

10-Regarding Fig 2a, and Fig S2a. Provide quantitation of these phenotypes. This is particularly important for the 3D culture, as it is unclear how the conclusions of less protrusive activity can be supported from the provided images. I can find in these alternate images alternate regions of interest to show different effects than those claimed. Without quantitation, these are selective and unscientific.

We now provide all the datapoints for the quantification of 2D data throughout the manuscript as detailed above. We also replaced the 3D culture data (steady state) from the first submission with a movie of MDA-MB-231 cells in 3D culture. (Supplementary Fig. 2c, Supplementary Movie 3). Our results show that cells continuously form long protrusions after ACM stimulation, and this process is diminished upon knockdown of CEP192. We were

unable to track cells in 3D for automated image analysis so we hope the movies we now include address this concern.

11-For statistical analyses throughout, there is a lack of consistency. The authors need to state to which conditions statistical analyses are compared. For example, in Fig 3a, presumably the control of statistical analysis is the second box plot, DMSO+ACM. In other figures, the first box plot is what conditions are compared to DMEM alone. Be consistent and provide a clear explanation in the main text as to why statistical analyses are not applied statistically in the main text body. Otherwise it is not clear between figures exactly what is being compared to what.

This is a good point. Overall, we use non-stimulated cells as control when we ask if there is a significant increase in motility (which is always the bar to the left); when we ask the question if there is a significant decrease/inhibition in cell motility we use stimulated cells as control, (which is always the second bar). For clarity, we now used an arrow in all graphs that run One-Way ANOVA analysis to indicate the control bar used for comparison.

12-Regarding Fig S3w, the position of the significance-indicating asterisks is mis-positioned, and hard to see.

We apologize but we are not sure what figure the referee is pointing to here since we did not have a Fig. S3w in the original manuscript. Nonetheless, we've gone through all the asterisks in all the figures, and hopefully the figure this reviewer had issues with has now been fixed. Otherwise, please let us know and we will fix it.

13-Regarding Fig S4, why is there a lack of bidirectionality in the LUMIER approach? For instance, PLK4 fails to fish out ARKB/C, but these fish out PLK4. Similarly, between PLK4 and CEP192? This is not mentioned.

Thank you very much for pointing it out, and it is a fair consideration. First of all, we noticed we made a labeling error in the heat map, although the NLIR values and labels in the Supplementary Table 1 excel file were correct. We've now verified the labeling of the heatmap, and the new heatmap shows PLK4 and CEP192 interacting with each other bidirectionally. As the reviewer mentioned, we also see AURKB(FLAG) pulling down CEP192 (*Renilla luciferase* (RL)), but CEP192(FLAG) did not pull down AURKB(RL). Here we used a high NLIR value (NLIR=6) as threshold to "call" interactions in order to focus on the strongest interactions. The CEP192(FLAG)-AURKB(RL) NLIR value is about 3, which we previously showed robustly detects significant interactions¹⁵, so this is significant, but not as strong. In addition, the *Renilla luciferase* used to tag the baits is relatively large, at 36KD and may affect interaction between bait and preys, especially for small baits such as AURKB that is only 39KD. Regardless, we validated the important interactions found in LUMIER by co-IP (Fig. 4 and Supplementary Fig. 4 and 5), and therefore we believe LUMIER is reasonable for high-throughput identification of novel protein-protein interactions.

14-Regarding Fig 4 and S4. There is a wide variation in input expression of a number of these exogenously expressed proteins. Provide quantitation of these experiments performed in

triplicate, so to ascertain whether these are really effects on stability, as claimed by authors, or whether these are simply experimental variation or transfection efficiency issues.

Thank you very much for pointing this out. This is the same issue pointed by Reviewer 1 (Minor point 2). We now provide the quantification data of the domain mapping of DVL2 with AURKB (Fig. R1-1c and 1d, Supplementary Fig. 5d), PLK4 (Fig. R1-1e and 1f, Supplementary Fig. 5e) and CEP192 (Supplementary Fig. 5c). Please see detailed response to this in Reviewer 1 (Minor point 2).

15-The authors describe (Fig 4) that Dvl2 seems to be controlling ‘steady-state’ levels of PLK4. However, all of these experiments are performed with exogenously expressed protein. Does Dvl2 depletion or overexpression change actual steady-state levels (i.e. endogenous levels) of PLK4 or AURKB/C? The authors present Dvl2 KD and OX, so it is curious why this is not provided.

We agree with the reviewer that investigating if changes in endogenous PLK4 levels correlate with DVL2 level is interesting. However, given that PLK4 is one of the least abundant centrosomal protein in cells²³, endogenous PLK4 is hard to detect and this has been a long standing problem for the field. Therefore, we could not detect reliable changes of PLK4 level upon DVL2 knockdown, which presumably would decrease PLK4 levels below already very low levels. However, by inducing DVL2 expression, our data shows that endogenous PLK4 levels increase (Supplementary Figs. 9b and 9c). Moreover, PLK4 colocalizes with DVL2 at protrusions (Supplementary Figs. 7d and 7g) and its level is increased in a DVL2-dependent manner after ACM stimulation (Figs. 7a and 7b). We therefore argue that DVL2 also contributes to the regulation of endogenous PLK4 stabilization, although the precise mechanism requires further investigations that are beyond the scope of this study.

16-Regarding the quantitation of cortical levels of protein, from Fig5 onwards. It is inappropriate to use a method of ‘cortical’ recruitment quantitation wherein the definition of the cortex is changed between conditions. Specifically, to compare the entire cortex in one condition, versus only a protrusion in another condition is unscientific and misleading. I understand that the authors are attempting to compare specialised functional zone (protrusion) versus unspecialised steady state (general cortex). However, these are not equivalent. Either compare sampling of the whole cortex in all conditions, or equally sized subzone vs equally sized subzone.

We agree with the reviewer that it is better to compare similarly sized regions. We have redone all the quantification as now described in Supplementary Fig. 6d and in Methods. Briefly, in cells that make protrusions, we selected the protrusions according to the actin structure as initially described. In cells that failed to make protrusions, we first dilated the nucleus (using DAPI) to remove the cortical regions that are close to the nucleus so that selected regions would be similarly distanced from the nucleus as that for protrusions. After removing these areas, the algorithm randomly picks two boxes per cell, which are of the same size as those used for selecting protrusions, from the remaining cortical region. Last, the average intensity of POI under these regions are measured, plotted and compared.

17-Regarding Fig 6g,h. The authors are positing that DVL2 inhibits BTRC binding to PLK4, thereby likely leading to PLK4 stabilisation. If this is the case, why does overexpressing lower and medium levels of exogenous Dvl2 initially stabilize the PLK4-BTRC interaction, and lead to increased PLK4 levels? It is only when very high levels DVL2 are expressed is the BTRC-PLK4 levels disrupted. Moreover, the levels of PLK4 appear the same, whether or not its association with BTRC is disrupted. The data do not therefore support such a simple model.

This is a good point and it is the same as pointed out by Reviewer 1 (Minor point 4). We believe this small increase may be due to a concomitant increase of overall PLK4 amount under low DVL2 expression. We cannot however, at this stage, exclude the possibility that other mechanisms may be involved in stabilizing the PLK4/BTRC interaction under moderate overexpression of DVL2. Please see detailed response above in Reviewer 1 (Minor point 4).

18-The discussion is overly long and mostly a rehash of the data. It would be very helpful to have a more concise summary of how the presented data fits in with the current literature, and the previous pathways identified by these groups.

We appreciate this suggestion, and we have now reduced the size of the discussion focusing on the implication of our findings relative to the field.

Figure R1-1

Figure R1-1. a-b, MDA-MB-231 cells stably expressing tetracyclin-inducible C-terminally 3Flag-tagged PLK4 or AURKB were incubated with or without 0.5 μ g/ml tetracycline overnight. **(a)** Cell lysates were processed for western blotting with anti-PLK4 (upper panel), anti-AURKB (lower panel) and anti-tubulin antibodies. Blue arrowheads indicate endogenous PLK4 or AURKB, red arrowheads indicate induced PLK4-3Flag or AURKB-3Flag. (N=3). **(b)** MDA-MB-231 cell lines characterized in **(a)** and **(b)** were stimulated with DMEM or ACM in the presence of centrinone or AZD as indicated. Cell motility was measured and plotted as box-and-whiskers. (****p<0. 0001; N=3, n>40). **c-f,** Association efficiency quantification for 3Flag-DVL2 wild type and domain mutants after co-immunoprecipitation with **(c)** 3HA-AURKB or **(e)** 3HA-PLK4 in HEK 293T cells. Total expression levels in whole cell lysates of 3Flag-DVL2 wild type and domain mutants when co-transfected with **(d)** 3HA-AURKB or **(f)** 3HA-PLK4. Quantification was derived from three independent experiments of which representative western blots are shown in **Supplementary Figure 5.**

Figure R2-1

a

b

c

Figure R2-1. a, MDA-MB-231 Cells were stimulated with DMEM or ACM for 16 h in the presence of DMSO or 0.5 $\mu\text{g/ml}$ Actinomycin D. Cell motility was measured and plotted as box-and-whiskers. (**** $p < 0.0001$; $N=3$, $n > 40$). **b,** MDA-MB-231 cells were stimulated with DMEM or ACM overnight. Cell motility was tracked for 3h, then cells were treated with DMSO or Actinomycin D and further tracked for 16h. Running average speed was plotted ($N=3$, $n > 40$). **c,** MCF10A cells were transiently transfected with GFP-PLK4 for 24 hours, followed by scratching a wound and cells were fixed 3h later in PFA. Cells were then stained for GFP and (a) phospho-PLK4 (S305), (b) CEP192 and (c) phospho-AURKB (T232). Left panel shows low exposure of image to display GFP-PLK4 at centrosome; while the right panel shows the same image with high exposure to display GFP-PLK4 and POI location in entire cell. Bar= $20 \mu\text{m}$. Arrow head indicates GFP-PLK4 positive cell, dash line indicates the wound.

Reference

1. Shnitsar I, *et al.* PTEN regulates cilia through Dishevelled. *Nat Commun* **6**, 8388 (2015).
2. Sonnen K, Gabryjonczyk A, Anselm E, Stierhof Y, Nigg E. Human Cep192 and Cep152 cooperate in Plk4 recruitment and centriole duplication. *J Cell Sci* **126**, 3223-3233 (2013).
3. O'Rourke B, *et al.* Cep192 controls the balance of centrosome and non-centrosomal microtubules during interphase. *PLoS One* **9**, e101001 (2014).
4. Fung E, *et al.* FBXL13 directs the proteolysis of CEP192 to regulate centrosome homeostasis and cell migration. *EMBO Rep* **19**, e44799 (2018).
5. Zhu F, *et al.* The mammalian SPD-2 ortholog Cep192 regulates centrosome biogenesis. *Curr Biol* **18**, 136-141 (2008).
6. Brownlee C, Klebba J, Buster D, Rogers G. The Protein Phosphatase 2A regulatory subunit Twins stabilizes Plk4 to induce centriole amplification. *J Cell Biol* **195**, 231-243 (2011).
7. Yokoyama N, Malbon C. Phosphoprotein phosphatase-2A docks to Dishevelled and counterregulates Wnt3a/beta-catenin signaling. *J Mol Signal* **2**, 12 (2007).
8. Rapchak C, Patel N, Hudson J, Crawford M. Developmental role of plk4 in *Xenopus laevis* and *Danio rerio*: implications for Seckel Syndrome. *Biochem Cell Biol* **93**, 396-404 (2015).
9. Fernandez-Miranda G, *et al.* Genetic disruption of aurora B uncovers an essential role for aurora C during early mammalian development. *Development* **138**, 2661-2672 (2011).
10. Gwee S, *et al.* Aurora kinase B regulates axonal outgrowth and regeneration in the spinal motor neurons of developing zebrafish. *Cell Mol Life Sci* **75**, 4269-4285 (2018).
11. Rosario C, *et al.* A novel role for Plk4 in regulating cell spreading and motility. *Oncogene* **34**, 3441-3451 (2015).
12. Kazazian K, *et al.* Plk4 Promotes Cancer Invasion and Metastasis through Arp2/3 Complex Regulation of the Actin Cytoskeleton. *Cancer Res* **77**, 434-447 (2017).
13. Gupta G, *et al.* A Dynamic Protein Interaction Landscape of the Human Centrosome-Cilium Interface. *Cell* **163**, 1484-1499 (2015).
14. Comartin D, *et al.* CEP120 and SPICE1 cooperate with CPAP in centriole elongation. *Curr Biol* **23**, 1360-1366 (2013).

15. Barrios-Rodiles M, Ellis J, Blencowe B, Wrana J. LUMIER: A Discovery Tool for Mammalian Protein Interaction Networks. *Methods Mol Biol* **1550**, (2017).
16. Barrios-Rodiles M, *et al.* High-throughput mapping of a dynamic signaling network in mammalian cells. *Science* **307**, 1621-1625 (2005).
17. Goodrich L, Strutt D. Principles of planar polarity in animal development. *Development* **138**, 1877-1892 (2011).
18. Sharma M, Castro-Piedras I, Simmons GJ, Pruitt K. Dishevelled: A masterful conductor of complex Wnt signals. *Cell Signal* **47**, 52-64 (2018).
19. Itoh K, Brott B, Bae G, Ratcliffe M, Sokol S. Nuclear localization is required for Dishevelled function in Wnt/beta-catenin signaling. *J Biol* **4**, 3 (2005).
20. Lee Y, *et al.* Dishevelled has a YAP nuclear export function in a tumor suppressor context-dependent manner. *Nat Commun* **9**, 2301 (2018).
21. Luga V, *et al.* Exosomes mediate stromal mobilization of autocrine Wnt-PCP signaling in breast cancer cell migration. *Cell* **151**, 1542-1556 (2012).
22. Zhang L, *et al.* A lateral signalling pathway coordinates shape volatility during cell migration. *Nat Commun* **7**, 11714 (2016).
23. Bauer M, Cubizolles F, Schmidt A, Nigg E. Quantitative analysis of human centrosome architecture by targeted proteomics and fluorescence imaging. *EMBO J* **35**, 2152-2166 (2016).

REVIEWERS' COMMENTS:

Reviewer #1 (Remarks to the Author):

The authors have addressed all of my comments. Although some issues were not clarified well for technical reasons, this reviewer is supportive for publication of the revised version in Nature Communications.

Reviewer #2 (Remarks to the Author):

The authors have addressed all my comments satisfactorily. I would support the addition of the actinomycin data to the supplementary results if space restrictions can be relaxed.

Reviewer #3 (Remarks to the Author):

All of my concerns have been addressed and I feel this paper is more than suitable for publication in Nature Communications. I feel that these authors have done an exemplary job responding to the plethora of reviewer comments.

Reviewer #4 (Remarks to the Author):

The revised manuscript by Luo et al is much improved and addresses all of my comments. A pleasure to read and review now.

REVIEWERS' COMMENTS:

Reviewer #1 (Remarks to the Author):

The authors have addressed all of my comments. Although some issues were not clarified well for technical reasons, this reviewer is supportive for publication of the revised version in Nature Communications.

We thank this reviewer for the insightful comments and suggestions. We are very pleased of his support to publish our revised manuscript in Nature Communications.

Reviewer #2 (Remarks to the Author):

The authors have addressed all my comments satisfactorily. I would support the addition of the actinomycin data to the supplementary results if space restrictions can be relaxed.

We are very pleased that this reviewer is satisfied with our revised version of the manuscript. We thank this reviewer for all the comments and suggestions, in particular the Actinomycin D experiment. We have now included the corresponding data as Supplementary Figure 3e and 3f, and it is mentioned in the text on Page 12.

Reviewer #3 (Remarks to the Author):

All of my concerns have been addressed and I feel this paper is more than suitable for publication in Nature Communications. I feel that these authors have done an exemplary job responding to the plethora of reviewer comments.

We thank this reviewer for eagerly supporting the publication of our revised manuscript in Nature Communications. We appreciate all the comments and suggestions for its improvement.

Reviewer #4 (Remarks to the Author):

The revised manuscript by Luo et al is much improved and addresses all of my comments. A pleasure to read and review now.

We are delighted that this reviewer enjoyed the revised version of the manuscript. We thank this reviewer for all the comments and suggestions that contributed to its improvement.